# A tRNA-derived fragment present in *E. coli* OMVs regulates host cell gene expression and proliferation

Idrissa Diallo[1], Jeffrey Ho[1], Marine Lambert[1], Abderrahim Benmoussa[1], Zeinab Husseini[1], David Lalaouna[2¤], Eric Massé[2], Patrick Provost[1]*

**1** CHU de Québec-Université Laval Research Center/CHUL Pavilion, Department of Microbiology, Infectious Diseases and Immunology, Faculty of Medicine, Université Laval, Quebec City, Quebec, Canada,
**2** CRCHUS, RNA Group, Department of Biochemistry and Functional Genomics, Faculty of Medicine and Health Sciences, Université de Sherbrooke, Sherbrooke, Quebec, Canada

¤ Current address: Université de Strasbourg, CNRS, Strasbourg, France
* patrick.provost@crchudequebec.ulaval.ca

**Data Availability Statement:** All raw small RNA-seq data generated in this study have been submitted to the NCBI Gene Expression Omnibus under accession number: BioProject accession: PRJNA826503; GEO accession: GSE200758 DOI :

## Abstract

RNA-sequencing has led to a spectacular increase in the repertoire of bacterial sRNAs and improved our understanding of their biological functions. Bacterial sRNAs have also been found in outer membrane vesicles (OMVs), raising questions about their potential involvement in bacteria-host relationship, but few studies have documented this issue. Recent RNA-Sequencing analyses of bacterial RNA unveiled the existence of abundant very small RNAs (vsRNAs) shorter than 16 nt. These especially include tRNA fragments (tRFs) that are selectively loaded in OMVs and are predicted to target host mRNAs. Here, in *Escherichia coli* (*E. coli*), we report the existence of an abundant vsRNA, Ile-tRF-5X, which is selectively modulated by environmental stress, while remaining unaffected by inhibition of transcription or translation. Ile-tRF-5X is released through OMVs and can be transferred to human HCT116 cells, where it promoted MAP3K4 expression. Our findings provide a novel perspective and paradigm on the existing symbiosis between bacteria and human cells.

## Author summary

We previously outlined by RNA-Sequencing (RNA-seq) the existence of abundant very small (<16 nt) bacterial and eukaryote RNA (vsRNA) population with potential regulatory functions. However, it is not exceptional to see vsRNA species removed from the RNA-seq libraries or datasets because being considered as random degradation products. As a proof of concept, we present in this study a 13 nt in length isoleucine tRNA-derived fragment (Ile-tRF-5X) which is selectively modulated by nutritional and thermal stress while remaining unaffected by transcription and translation inhibitions. We also showed that OMVs and their Ile-tRF-5X vsRNAs are delivered into human HCT116 cells and both can promote host cell gene expression and proliferation. Ile-tRF-5X appears to regulate gene silencing properties of miRNAs by competition. Our findings provide a novel perspective and paradigm on the existing symbiosis between hosts and bacteria but also

https://www.ncbi.nlm.nih.gov/geo/query/acc.cgi?acc=GSE200758.

**Funding:** This work, and the article processing charge (APC) were funded by Natural Sciences and Engineering Research Council of Canada (NSERC; https://www.nserc-crsng.gc.ca) Discovery Grant number RGPIN-2019-06502 (to P.P.). The funders had no role in study design, data collection and analysis, decision to publish, or preparation of the manuscript.

**Competing interests:** The authors have declared that no competing interests exist.

brings a new insight of host-pathogen interactions mediated by tRFs which remain so far poorly characterized in bacteria.

## Introduction

High-throughput sequencing (HTS) techniques have led to an explosion of knowledge on non-coding RNAs (ncRNAs) [1] and, more importantly, they have shed light on the extent to which they are involved in the regulation of gene expression in both prokaryotes and eukaryotes [2–4]. In bacteria, ncRNAs are generally referred to as small non-coding RNAs (sRNAs). The regulatory role of bacterial sRNAs was first demonstrated in extrachromosomal systems [5] and then fortuitously in intrachromosomal systems, in which the sRNA MicF significantly inhibited translation of the *ompF* gene through binding to its 5' end [6].

Most of the sRNAs rely on base pairing interactions, which in eukaryotes may involve active protein complexes [7, 8] leading to translation blockade and the degradation or stabilization of the target messenger RNA (mRNA) [9]. sRNA:mRNA interactions may also lead to transcriptional or translational activation [10, 11].

sRNAs originate from multiple sources other than the intergenic regions of bacterial genomes [12]. Moreover, the 3' untranslated region (UTR) of mRNAs emerged as a potential major reservoir of sRNAs [13–15]. Barquist and colleagues [16] have also introduced original evolutionary concepts on transcriptional noise and exaptation as sources of bacterial sRNAs.

With the unprecedented pace of technical advances and sRNA discovery [17, 18], the paradigm of their origin has been extended to two major classes of ncRNAs: rRNAs and tRNAs. Studies in eukaryotes revealed that rRNA fragments (rRFs) and tRNA fragments (tRFs) are not degradation products, but rather functional sRNAs with specific expression patterns and functions [19–22]. Lalaouna et al., [23] have also evidenced a functional bacterial tRF corresponding to an external transcribed spacer (ETS) which acts as a sponge for sRNAs to prevent transcriptional noise.

However, the literature on the existence, characteristics, biogenesis or function of tRFs in bacteria is scarce; the nomenclature not well established, and little is known about their potential role in microbial physiology and host-bacteria relationships, which is likely underappreciated [24]. The biogenesis of tRFs, their cell-autonomous effect, their transfer to host cells and the emergence of bacterial sRNAs as virulence factors in host-bacteria interactions were reviewed recently [25–27].

In bacteria-host interactions that occur in large interface areas, such as the colon, where the bacterial load exceeds other organs by two orders of magnitude [28], it is assumed that the microbiome employs immunostimulatory mechanisms capable of activating the host immune system and afferent pathways [29]. However, these interactions are often seen from the perspective of macromolecules [30], overlooking *de facto* the potential role of small RNAs and that of strategic carriers like outer membrane vesicles (OMVs).

Bacterial extracellular vesicles such as OMVs (50–250 nm) are spherical particles produced and released in all domains of life. These extracellular particles are rich in active biomolecules and perform multiple intercellular functions [31]. OMVs also constitute an attractive "type zero" secretion system capable of delivering their contents to the host without directly exposing the bacteria from which they originate [32]. The RNA content of bacterial OMVs has been long hypothesized [33] and was established much later [34–36] than that of extracellular vesicles derived from eukaryotic cells [37, 38]. The RNA of extracellular vesicles may have the potential to shape microbial communities and host–microbe interactions [39].

Beyond their profile, little is known about the biological role of bacterial RNAs once transferred into human host cells. Recent studies have described the first example of trans-kingdom biological activity of regulatory sRNAs contained in OMVs derived from *Pseudomonas aeruginosa* reference strain PA14 [40], *Legionella pneumophila* strain Paris [41] and periodontal pathogens [42].

Our group recently reported the discovery of abundant, unusually short (12–13 nt) and functional dodecaRNAs (doRNAs) mapping to rRNA 5.8S in eukaryotes [43]. In line with this work, we identified and characterized very small RNAs (vsRNAs) shorter than 16 nt using RNA-Sequencing (RNA-Seq) in *E. coli* and five other bacterial strains (*Pseudomonas aeruginosa* PA7, *P. aeruginosa* PAO1, *Salmonella enterica* serovar Typhimurium 14028S, *L. pneumophila* JR32 Philadelphia-1 and *Staphylococcus aureus* HG001). These vsRNAs were highly abundant in *E. coli* and their derived OMVs, with heterogeneous and distinct populations. The loading of *E. coli* vsRNAs, especially tRFs, in OMVs seemed to be selective. We also observed that tRFs are probably produced upon specific processing of tRNA species, form thermodynamically stable hairpin structures, and are predicted to target several host mRNAs with diverse functions [44].

We hypothesized that the dynamic host-gut microbiota interaction may involve an exchange mechanism based on OMVs and implicating the transfer of bacterial tRFs to human host cells, with possible implications in pathological and/or homeostasis processes. To test this hypothesis, we studied a tRF belonging to the 5' end of the mature Isoleucine (Ile)-tRNA (tRF-5, **Fig 1**), which we found to be the most abundant tRF in both *E. coli* and their OMVs. The tRF-5s are divided into subclasses: tRF-5a (14–16 nts), tRF-5b (22–24 nts) and tRF-5c (28–30 nts) [45]. Our tRF-5 of interest being 13-nt long was tentatively named Ile-tRF-5X.

Here, we report that bacterial Ile-tRF-5X can be selectively modulated by environmental stress and, once delivered through OMVs to human host HCT116 cells, can regulate expression of a component of the mitogen-activated protein kinases (MAPKs) pathway through competition with miRNA silencing. We also show that OMVs and Ile-tRF-5X enhance proliferation of cultured colorectal carcinoma cells.

## Results

### Ile-tRF-5X levels in *E. coli* MG1655 are increased upon nutritional and thermal stresses

Before studying vsRNAs in the bacteria-host relationship, we characterized the level of vsRNA Ile-tRF-5X in reference strain *E. coli* MG1655 under different growth conditions. To this end, we designed and used a sensitive and specific RT-qPCR detection method based on splint ligation to monitor Ile-tRF-5X level. The technique is detailed in the Materials and methods section and illustrated in **S1A–S1C Fig**.

First, we evaluated the level of Ile-tRF-5X expression at the exponential and stationary phases of bacterial growth (**Fig 2A**). In a rich and complete LB medium, although the mature Ile-tRNA appeared to accumulate in the stationary phase (about 3-fold more than Ile-tRF-5X; **S2 Fig**), Ile-tRF-5X level seemed not to be affected by the growth phase, as suggested also by our RNA-Seq data (see **S3 Fig**).

However, in low nutrient culture medium (M63), the level of Ile-tRF-5X was significantly increased in both exponential (by 2-fold) and stationary (by 4-fold) phases compared to exponential growth phase in LB medium, chosen as a reference (**Fig 2A**). Under these respective conditions, the Ile-tRNA was reduced by 20 (3-fold less than Ile-tRF-5X) and 60% (11-fold less than Ile-tRF-5X) compared to the reference (**S2 Fig**).

Next, we investigated the level of Ile-tRF-5X in function of temperature (**Fig 2B**) at 30˚C and 44˚C, in comparison with exponential growth at the optimal reference temperature of 37˚C. Compared to the reference, the level of Ile-tRF-5X significantly increased by 2-fold

**Fig 1. Simplified organization of the 3 operons containing the Ile-tRNA source of Ile-tRF-5X.** *isoleucine*(Ile)-tRNA (GAT = anticodon) and *alanine*(Ala)-tRNA (TGC = anticodon) genes are found in the rrnA, rrnD and rrnH operons of *E. coli* K-12 genome. The Ile and Ala tRNAs are flanked by the 16S and 23s ribosomal RNAs (rRNAs). *ileT*, *ileU*, *ileV* genes all produce Ile(GAU)-tRNA. *alaT*, *alaU*, *alaV* genes all produce Ala(UGC)-tRNA. This diagram represents the three transcription units in one panel, under which the genomic sequence of Ile-tRNA and its mature sequence are presented. The 13-nt sequence in red represents Ile-tRF-5X, with its predicted secondary structure. The diagram does not show the endo and exonucleases involved in the maturation at each stage. The ribonuclease(s) involved in the cleavage of Ile-tRF-5X 3' end have yet to be identified. Str, strain; substr, substrain; tRNA, transfer RNA.

when raising the temperature to 44˚C (+7˚C), but remained unchanged when the temperature was lowered to 30˚C (-7˚C). The level of mature Ile tRNA followed the same pattern as Ile-tRF-5X at both temperatures tested (**S2 Fig**)

Therefore, the level of Ile-tRF-5X appears to be modulated by two major growth conditions (i.e., nutrients availability and temperature) encountered by bacteria.

## Ile-tRF-5X is not affected by inhibition of transcription or translation

We then assessed whether the level of Ile-tRF-5X could be related to transcription or translation, for which we have used rifampicin and chloramphenicol, respectively. Rifampicin inhibits the bacterial RNA polymerase (RNAP) and stops global RNA synthesis at the initiation of transcription, while having no effect on replication elongation [46]. Chloramphenicol, on the other hand, has no effect on transcription and rather targets the 50S ribosomal subunit and inhibits peptidyl transferase activity, resulting in the cessation of protein synthesis [47].

Inhibition of RNA and protein synthesis, with rifampicin and chloramphenicol respectively, did not affect the level of bacterial Ile-tRF-5X (**Fig 3**). These RT-qPCR results confirm our RNA-Seq data suggesting that the level of bacterial Ile-tRF-5X is not affected by inhibition of transcription or translation (**S3 Fig**).

A complementary RT-qPCR analysis (**S2 Fig**) showed that the level of Ile-tRNA was affected by inhibition of RNA (20% decrease) and protein synthesis (70% increase), suggesting a lack of correlation between the level of tRNAs and that of tRFs.

## Regulation of Ile-tRF-5X levels may involve RNase E, but not RNase P

The intriguing invariability of Ile-tRF-5X levels following inhibition of transcription or translation led us to scrutinize its biosynthesis. To do so, the level of Ile-tRF-5X expression was assessed following inhibition of some key enzymes of the tRNA maturation pathway (**Fig 4**).

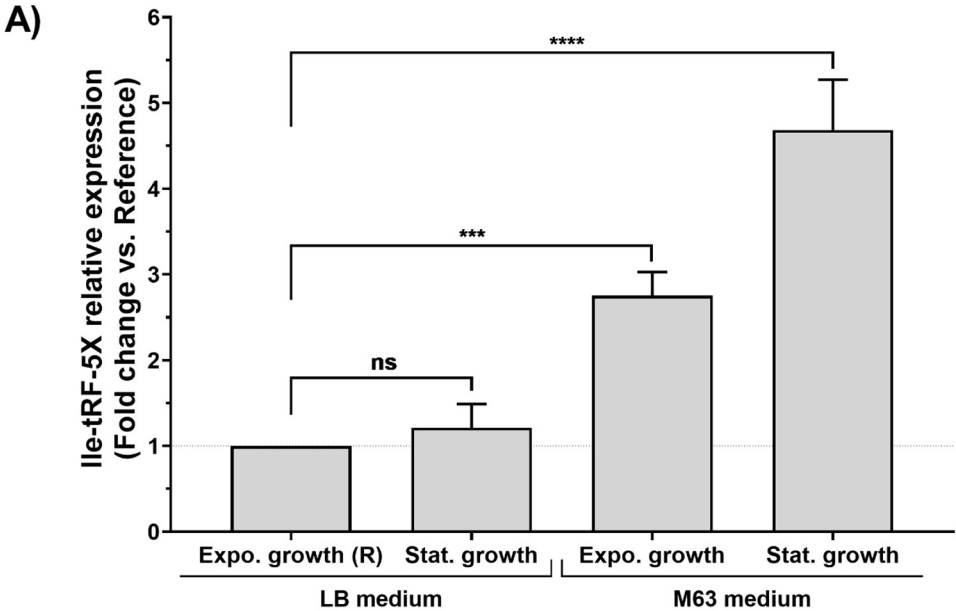

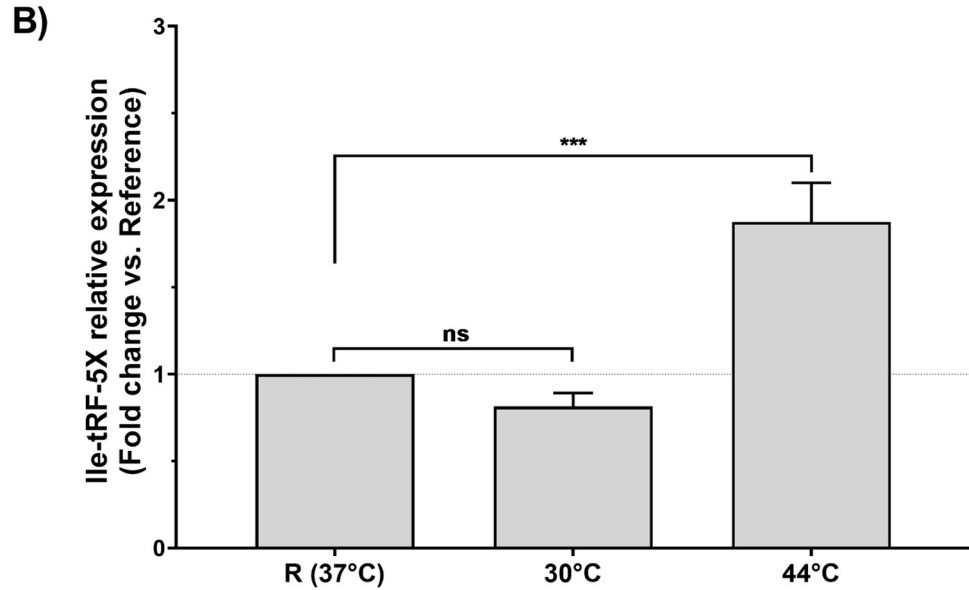

**Fig 2. Bacterial Ile-tRF-5X levels under different growth and temperature conditions.** (**A**) *E. coli* MG1655 bacteria were grown at 37˚C, and the level of Ile-tRF-5X was monitored during the exponential and stationary phases of growth in either complete (rich) LB or minimal M63 medium. (**B**) *E. coli* MG1655 bacteria were grown in LB medium, and the level of Ile-tRF-5X was monitored during the exponential phase at 30, 37 and or 44˚C (for details, see **S1 Table** and **S2 Fig**). The level of Ile-tRF-5X was measured by LNA RT-qPCR. A spike-in (UniSp6) and reference genes (23S and/or 16S) were used as control and for normalization. The results are reported in fold change compared to the reference condition. **Statistical analysis**. Data were calculated from three biological replicate measurements (n = 3; mean ± SD). One-way analysis of variance (ANOVA) and Holm-Šídák's multiple comparisons test (post-hoc test) were used for statistical analysis. Statistically significant differences (fold change vs. reference) are indicated by stars (*), * $p < 0.05$; ** $p < 0.01$; *** $p < 0.001$; **** $p < 0.0001$; ns, not significant. In the M63 medium, the difference between the exponential and stationary phase was statistically significant (***$p = 0.0006$).

tRNA processing involves several ribonucleases, such as RNase E and RNase P [48]. Involved in almost all aspects of RNA metabolism, RNase E initiates tRNA maturation and provides substrates to RNase P and other exoribonucleases [49]. RNase P, which exists in all three kingdoms of life, generates the mature 5' ends of tRNAs [50].

Transient and thermosensitive inhibition of RNase E at 44°C decreased the level of Ile-tRF-5X by half compared to the reference condition, as suggested by our RNA-Seq data (**S3 Fig**). These findings imply an active participation of RNase E in the biosynthesis of vsRNA Ile-tRF-5X. However, inhibition of RNase E did not lower the level of mature Ile-tRNA from which Ile-tRF-5X originated. Ile-tRNA was 30% more abundant than the control and 3-fold more abundant than Ile-tRF-5X (**S2 Fig**).

RNase P, on the other hand, does not seem to be involved in the processing of Ile-tRF-5X, as inhibition of this key tRNA maturation enzyme did not alter Ile-tRF-5X expression. This result leaves open the possibility that Ile-tRF-5X may originate from the mature and/or precursor tRNA. Unlike RNase E, the absence of RNase P did not significantly remodulate the level of Ile-tRNA (**S2 Fig**).

Taken together, these results suggest that Ile-tRF-5X is not only modulated by growth conditions, but also by general RNA metabolism mediated by RNase E. Ile-tRF-5X might therefore be considered as a product of a specific process.

## Bacterial OMVs are internalized by HCT116 cells

We have previously shown that a group of thermodynamically stable bacterial vsRNAs, including Ile-tRF-5X (**Fig 1**), were selectively enriched and loaded into OMVs ([44]; **S4 Fig**), suggesting a potential role in bacteria-host cell communications.

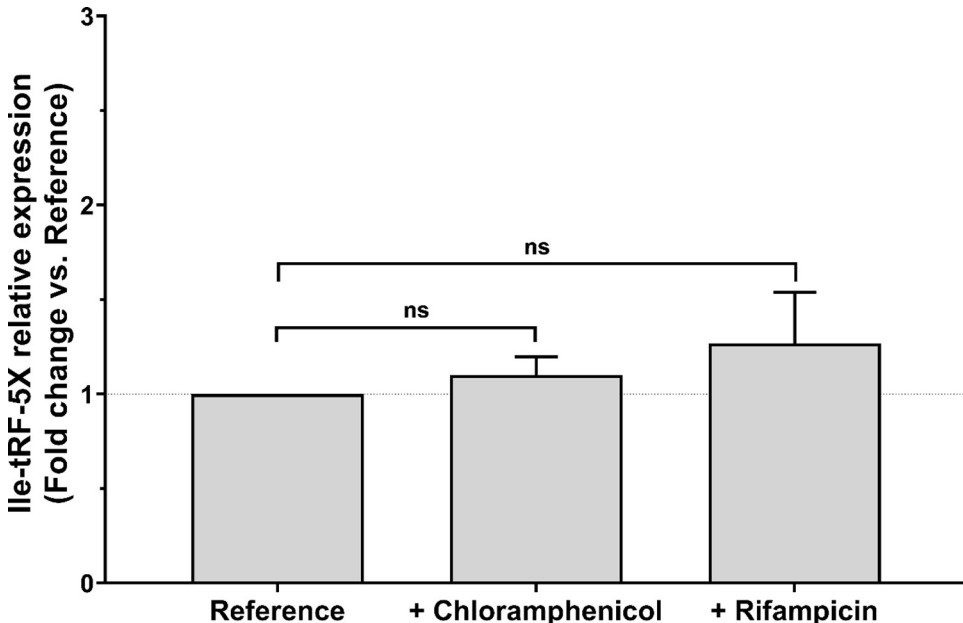

**Fig 3. The level of bacterial Ile-tRF-5X level is not modulated by transcription or translation activity.** Bacterial mRNA or protein synthesis was inhibited by addition of chloramphenicol or rifampicin, respectively, to cultures of *E. coli* MG1655 at 37°C (for details, see **S1 Table** and **S2 Fig**). The level of Ile-tRF-5X was measured by LNA RT-qPCR. A spike-in (UniSp6) and reference genes (23S and/or 16S) were used as control and for normalization. The results are reported in fold change compared to the reference condition. **Statistical analysis**. Data were calculated from three biological replicate measurements (n = 3; mean ± SD). One-way analysis of variance (ANOVA) and Holm-Šídák's multiple comparisons test (post-hoc test) were used for statistical analysis. Statistically significant differences (fold change vs. reference) are indicated by stars (*), * p < 0.05; ** p < 0.01; *** p < 0.001; **** p < 0.0001; ns, not significant.

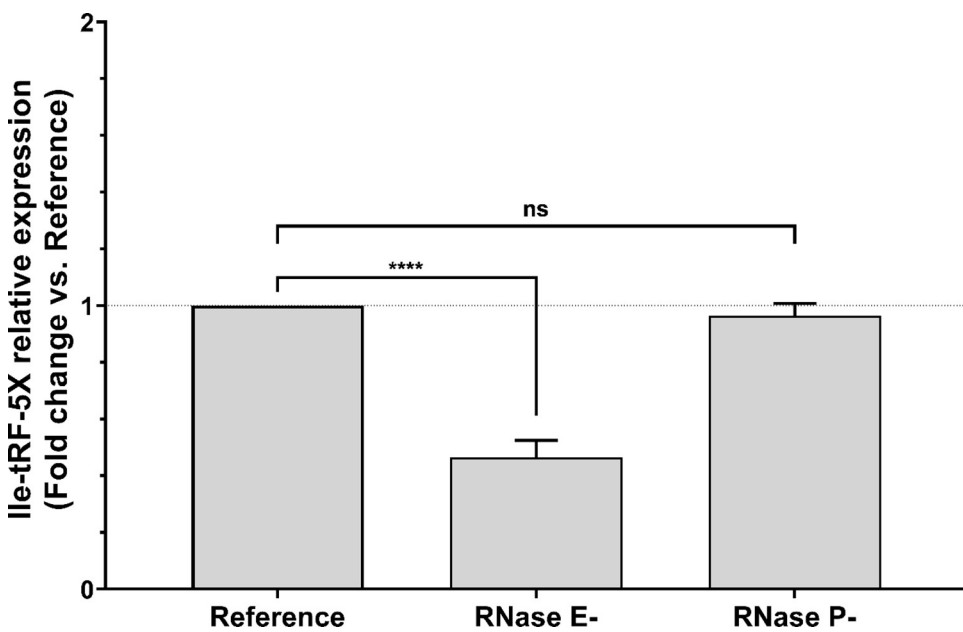

**Fig 4. Bacterial RNase E contributes to Ile-tRF-5X biogenesis.** *E. coli* strains carrying heat-sensitive (hs) mutations in the essential genes *rne-3071-hs* (EM1277) and *rnpA-hs* (KP1036) were grown in LB medium at 30˚C and then heat-shocked to transiently inhibit ribonuclease (RNase) P or RNase E, which are involved in tRNA maturation (for details, see **S1 Table** and **S2 Fig**). The level of Ile-tRF-5X was measured by LNA RT-qPCR. A spike-in (UniSp6) and reference genes (23S and/or 16S) were used as control and for normalization. The results are reported in fold change compared to the reference condition. **Statistical analysis**. Data were calculated from three biological replicate measurements (n = 3; mean ± SD). One-way analysis of variance (ANOVA) and Holm-Šídák's multiple comparisons test (post-hoc test) were used for statistical analysis. Statistically significant differences (fold change vs. reference) are indicated by stars (*), * $p < 0.05$; ** $p < 0.01$; *** $p < 0.001$; **** $p < 0.0001$; ns, not significant.

To investigate that possibility, we performed a confocal fluorescence microscopy analysis of labeled human colorectal carcinoma cell line HCT116 incubated with PKH67-labeled bacterial OMVs (**Fig 5**). Volocity's XZ and XZ projection features were used to verify the localization (membrane, cytoplasm, nuclei, etc.) of fluorophores (**S5 Fig**).

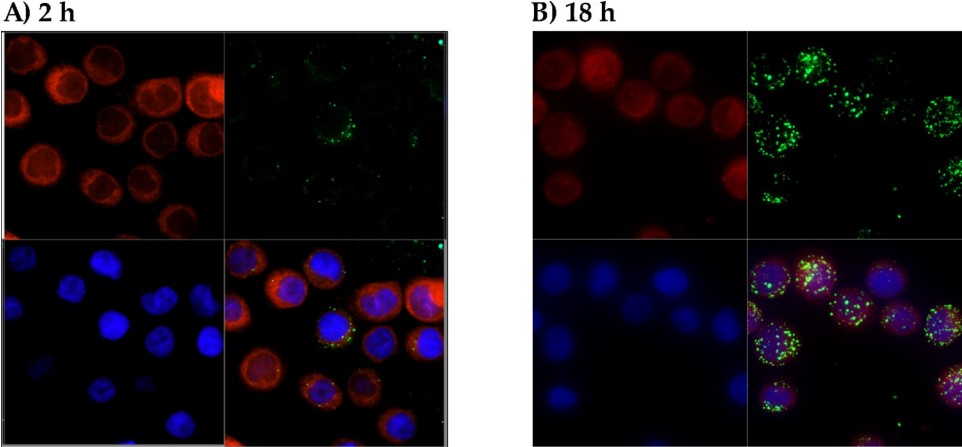

**Fig 5. Human HCT116 cells internalize bacterial OMVs.** Confocal microscopy imaging of human HCT116 cells stained with cell tracker CMTPX (in red) and incubated with bacterial OMVs labelled with PKH67 (in green) for 2 h (**A**) or 18 h (**B**). Nuclei were stained with DAPI (in blue). The XZ and YZ projection of volocity shows that bacterial OMVs were internalized by HCT116 cells and localized mainly to the cytoplasm (see **S5 Fig**).

After 2 h of incubation, a small proportion of bacterial OMVs were already found inside the cells, while the majority were still located on the cell surface. After 18 h of incubation, most of the bacterial OMVs were internalized and scattered in the cytoplasm. Some cells did not take up bacterial OMVs, suggesting a selective adhesion and internalization process.

## Bacterial Ile-tRF-5X is transferred to human HCT116 cells

After demonstrating that bacterial OMVs could be internalized by human host cells, we verified whether their content, especially Ile-tRF-5X, which is the most abundant vsRNA, could be transferred to host cells.

We monitored, by LNA RT-qPCR, the level of bacterial Ile-tRF-5X in HCT116 cells upon exposure to bacterial OMVs or synthetic Ile-tRF-5X (**Fig 6**).

HCT116 cells exposed to OMVs displayed high levels of Ile-tRF-5X at 24 h, which dropped by a third at 48 h, but remained significant. Cells transfected with synthetic Ile-tRF-5X displayed a marked increase in Ile-tRF-5X levels, compared to mock-transfected cells, which were two times higher than cells exposed to OMVs at 24 h. At 48 h, the level of Ile-tRF-5X dropped drastically by 74% compared to its level at 24 h. At 48 h, a third more Ile-tRF-5X was found in cells incubated with OMVs compared to those transfected with synthetic Ile-tRF-5X.

These results indicate that bacterial OMVs and their Ile-tRF-5X content can be transferred to human host cells.

## Bacterial OMVs promote MAP3K4 expression

Whether bacterial Ile-tRF-5X can function in human host cells and regulate host gene expression was initially explored using bioinformatic tools, which brought to light several host mRNAs that might be targeted by tRF vsRNAs, like Ile-tRF-5X [44]. Envisioning these interactions on large host-bacteria exchange surfaces such as the colon, we focused on related targets previously reported in this context and related to extracellular signal transduction,

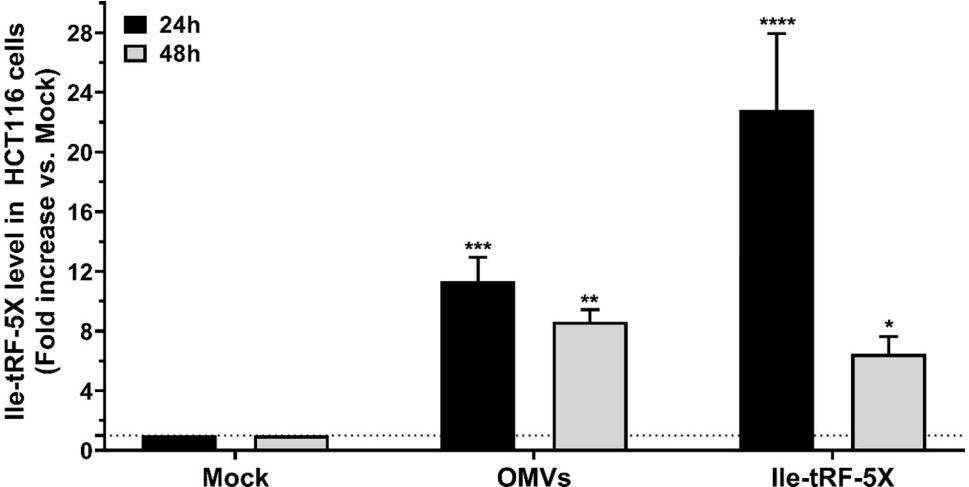

**Fig 6. Bacterial OMVs deliver Ile-tRF-5X to human HCT116 cells.** Human HCT116 cells were incubated with (10 ng) *E. coli* MG1655-derived OMVs or transfected with an equivalent amount of synthetic Ile-tRF-5X. The level of Ile-tRF-5X was measured at 24 h and 48 h by LNA RT-qPCR. UniSp6 was used as spike-in control and reference gene for normalization. The results were reported in fold change compared to the corresponding mock condition. **Statistical analysis.** Data were calculated from three biological replicate measurements (n = 3; mean ± SD). One-way analysis of variance (ANOVA) and Holm-Šídák's multiple comparisons test (post-hoc test) were used for statistical analysis. Statistically significant differences is indicated as follows: * $p < 0.05$; ** $p < 0.01$; *** $p < 0.001$; **** $p < 0.0001$.

inflammatory cytokines, or environmental stress. For further validation, and as a proof of concept of the functionality of Ile-tRF-5X, we retained Mitogen-activated protein kinase 3 mRNA (MAP3K4, also known as MEKK4), which carries no less than 10 potential binding sites, including one with perfect complementarity to Ile-tRF-5X located in the open reading frame (ORF) of MAPK3 (S6 Fig). Involved in several aspects of cellular regulation, MAPKs are important signal transducing enzymes which allow the transfer of signals from cell surface receptors to critical regulatory targets within cells [51].

We first exposed HCT116 cells to bacterial OMVs and measured the level of MAP3K4 expression. As OMVs naturally contain pathogen-associated molecular patterns (PAMPs), among which lipopolysaccharide (LPS) remains one of the major components, LPS was used as a control. The results showed that bacterial OMVs enhanced MAP3K4 expression at both messenger RNA and protein levels (Fig 7). Incubation of HCT116 cells with LPS did not alter MAP3K4 expression.

## Bacterial Ile-tRF-5X induces sequence-specific upregulation of MAP3K4 mRNA expression

To ascertain that the observed changes in MAP3K4 levels upon exposure to bacterial OMVs can be attributed to Ile-tRF-5X and not to other OMV components, we transfected HCT116 cells with either Ile-tRF-5X alone, a control sequence of the length of Ile-tRF-5X or small RNA (<200 nt) isolated from OMVs.

Ile-tRF-5X seemed to upregulate MAP3K4 only at the mRNA level (Fig 8A and 8B).

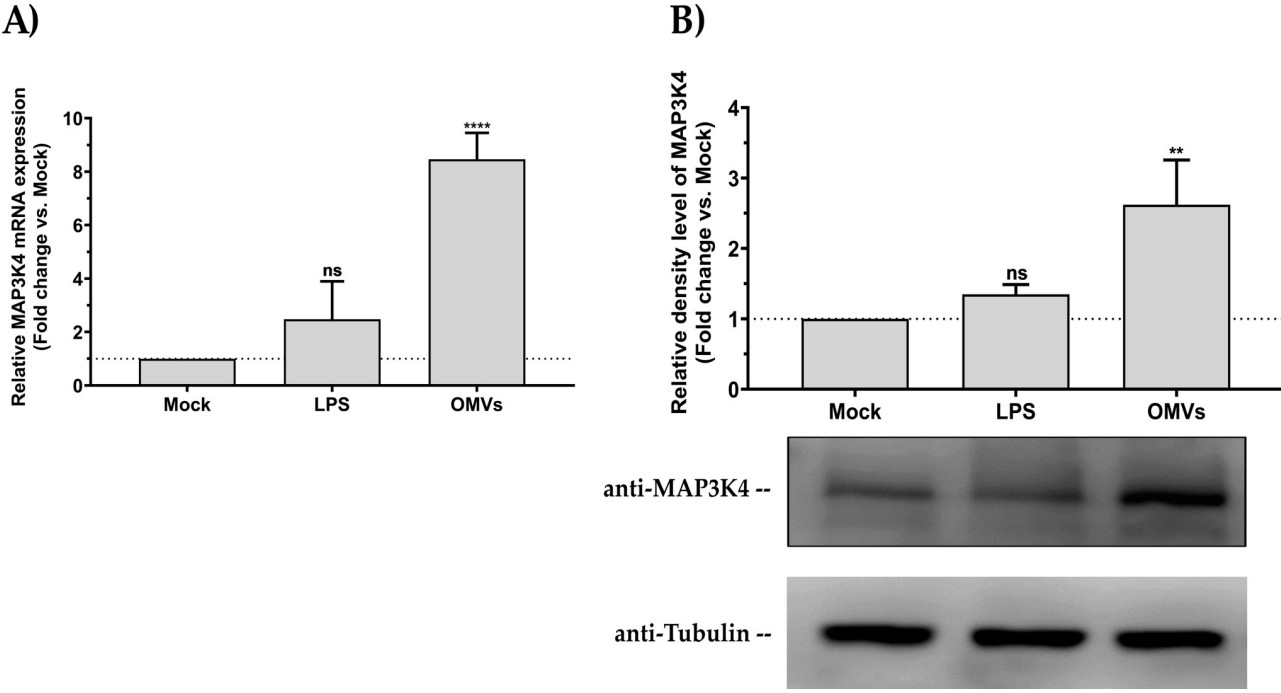

**Fig 7. Bacterial OMVs enhance human MAP3K4 expression.** Human HCT116 cells were incubated with (10 ng) *E. coli* MG1655-derived OMVs or an equivalent amount of LPS, compared to a mock control. (**A**) Relative quantification of MAP3K4 mRNA expression by RT-qPCR. Data were normalized with a reference gene (ACTB), reported as fold change vs mock control, and expressed with the relative quantitation method (ΔΔCt). (**B**) Western blot analysis of MAP3K4 protein expression relative to tubulin. Densitometric analysis of the bands was performed with ImageJ. Data were normalized with a reference gene (tubulin) and expressed as fold change vs mock control. **Statistical analysis.** Data were calculated from three biological replicate measurements (n = 3; mean ± SD). One-way analysis of variance (ANOVA) and Holm-Šídák's multiple comparisons test (post-hoc test) were used for statistical analysis. Statistically significant differences (fold change vs mock) are indicated as follows: ** $p < 0.01$; **** $p < 0.0001$. ns, not significant.

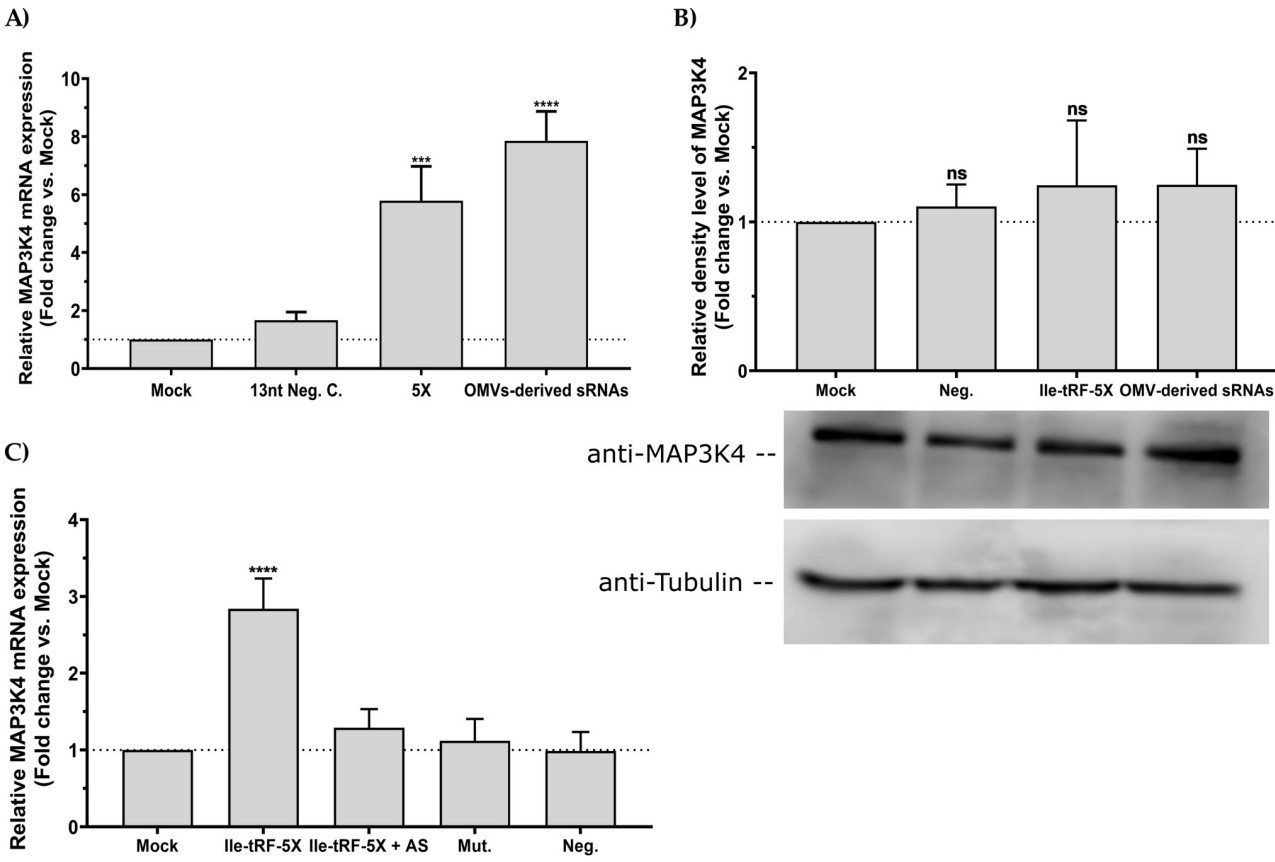

**Fig 8. Bacterial Ile-tRF-5X upregulates human MAP3K4 expression in a sequence specific manner.** (**A-C**) Human HCT116 cells were transfected with (100 nM) synthetic Ile-tRF-5X alone, mutated (Mut.) or combined with its antisense (AS), unrelated 13-nt negative control (Neg.) or small RNAs (sRNAs) derived from bacterial OMVs. (**A**) and (**C**) Relative quantification of MAP3K4 mRNA expression by RT-qPCR. Data were normalized with a reference gene (ACTB), reported as fold change vs mock control, and expressed with the relative quantitation method (ΔΔCt). (**B**) Western blot analysis of MAP3K4 protein expression relative to tubulin. Densitometric analysis of the bands was performed with ImageJ. Data were normalized with a reference gene (tubulin) and expressed as fold change vs mock control. **Statistical analysis.** Data were calculated from three biological replicate measurements (n = 3; mean ± SD). One-way analysis of variance (ANOVA) and Holm-Šídák's multiple comparisons test (post hoc test) were used for statistical analysis. Statistically significant differences are indicated as follows: *** $p < 0.001$, **** $p < 0.0001$. ns, not significant.

Interestingly, the cocktail of small OMV-derived RNAs (<200 nt) also appeared to modulate MAP3K4 expression, slightly more than Ile-tRF-5X alone.

To address the specificity of the interplay between Ile-tRF-5X and MAP3K4, a mutated version of Ile-tRF-5X was also tested as well as an antisense to Ile-tRF-5X. Mutated Ile-tRF-5X had no effect and the antisense neutralized the MAP3K4 gene regulatory effects of Ile-tRF-5X, supporting the sequence-specificity of the interaction between Ile-tRF-5X with MAP3K4 mRNA (**Fig 8C**).

## Bacterial OMVs and Ile-tRF-5X modulate human MAP3K4 expression through the miRNA pathway

The presence of putative binding sites to Ile-tRF-5X in MAP3K4 mRNA (**S6 Fig**) led us to speculate that bacterial vsRNAs may function like eukaryotic miRNAs or tRFs, i.e. through binding and downregulation of their mRNA targets [52]. However, the opposite upregulation of MAP3K4 levels induced by Ile-tRF-5X suggests that it might not follow this mechanism of action. Instead, we purported that vsRNAs might compete with human host miRNAs and/or tRFs involved in MAP3K4 gene regulation.

Both tRFs and miRNAs largely depend on the ribonuclease Dicer for their production, but the high variability of the mechanisms involved in tRF generation led us to focus on miRNAs. To probe the modulation of MAP3K4 expression, we thus used a Dicer-deleted (Dicer -/-) HCT116 cell line, which exhibit a 96% depletion of mature miRNA levels [53], in parallel to the wild-type HCT116 cell line.

First, we confirmed the depletion of human Dicer at the mRNA and protein levels in Dicer -/- HCT116 cells, compared to wild-type cells (**Fig 9A**). These cells were then incubated in the absence or presence of bacterial OMVs or transfected or not with synthetic Ile-tRF-5X, followed by measurement of MAP3K4 mRNA expression. In HCT116 WT, either with OMVs or Ile-tRF-5X alone,

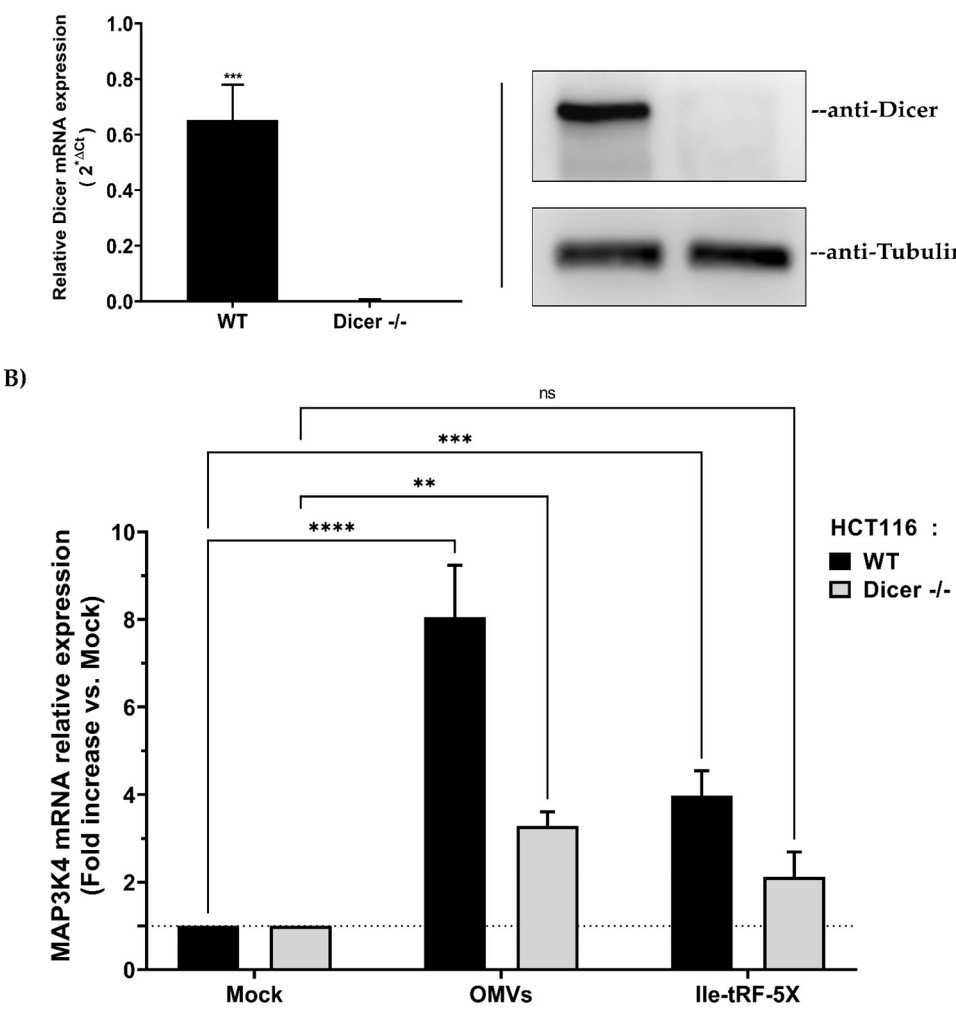

**Fig 9. Bacterial OMVs and Ile-tRF-5X modulate human MAP3K4 expression through the miRNA pathway.** (**A**) Dicer deletion (Dicer -/-) in human HCT116 cells was confirmed at the mRNA level by qPCR and at the protein level by Western blot using anti-Dicer and anti-Tubulin (control) antibodies. (**B**) Relative MAP3K4 mRNA expression was quantified at 48h by RT-qPCR in either wild-type (WT) or Dicer -/- HCT116 cells after incubation with bacterial OMVs (10 ng) or transfection with Ile-tRF-5X (100 nM). Data were normalized with a reference gene (ACTB), reported as fold change vs mock control, and expressed with the relative quantitation method (ΔΔCt). **Statistical analysis**. Data were calculated from three biological replicate measurements (n = 3; mean ± SD). Unpaired Student's t test (**A**) and two-way analysis of variance (ANOVA) and Holm-Šídák's multiple comparisons test (**B**) were used for statistical analysis. Statistically significant differences (fold change vs. mock) are indicated as follows: * $p < 0.05$; *** $p < 0.001$; **** $p < 0.0001$.

there was an increase in the level of MAP3K4 whereas in the experimental condition where Dicer was absent, only OMVs induced a significant upregulation of MAP3K4 (**Fig 9B**). The respective 8- and 4-fold increase in MAP3K4 mRNA expression induced by bacterial OMVs or Ile-tRF-5X alone in wild-type HCT116 cells was reduced by half upon Dicer depletion (**Fig 9B**).

These results suggest that the MAP3K4 gene regulatory effects of bacterial OMVs or Ile-tRF-5X are, at least partially, mediated by Dicer, either directly (Dicer itself) or indirectly, through its miRNA products.

## Bacterial Ile-tRF-5X counteracts the miRNA-induced gene downregulatory properties of MAP3K4 3'UTR

As the regulatory elements recognized by miRNAs are usually located in mRNA 3'UTRs [54], we inspected the MAP3K4 mRNA 3'UTR for miRNA binding sites, or miRNA response elements (MREs), using TargetScan [55] and for the five most important potential Ile-tRF-5X binding sites using RNAhybrid [56].

We observed a close proximity between the MREs and those of Ile-tRF-5X (**S7 Fig**).

At least two of the MREs in MAP3K4 mRNA 3'UTR overlap with Ile-tRF-5X binding sites (at nt positions 77 and 208). The other Ile-tRF-5X binding sites were located within 30 nt of MREs, with one within 3 nt (nt position 121). The secondary structure of MAP3K4 mRNA 3'UTR, depicted by the Forna (RNAfolder) web interface [57], revealed that Ile-tRF-5X binding sites were mainly located in structured sequences, often in stem loops that made them even closer to MREs (**S8 Fig**).

To study the possible interactions between miRNAs and Ile-tRF-5X in the regulation of MAP3K4 mRNA, we cloned the entire MAP3K4 mRNA 3'UTR into a dual-luciferase reporter gene system, which was used in wild-type and Dicer -/- HCT116 cells (**Fig 10**).

Introduction of the MAP3K4 mRNA 3'UTR downstream of Rluc reduced reporter gene expression in wild-type HCT116 cells by 60% compared to the empty vector (in the absence of added Ile-tRF-5X; **Fig 10A**), suggesting the presence of gene downregulatory elements in the 3'UTR. Addition of Ile-tRF-5X restored Rluc activity in a dose-dependent manner, supporting the ability of Ile-tRF-5X to counteract the gene downregulatory factors acting via the MAP3K4 mRNA 3'UTR (**Fig 10A**).

To determine the specificity of the Ile-tRF-5X gene regulatory effects, we performed the same experiment with mutated Ile-tRF-5X binding sites, without altering MREs. Rluc reporter gene activity was reduced by only 35% (**Fig 10B**), compared to as much as 60% for the wild-type MAP3K4 mRNA 3'UTR (**Fig 10A**), suggesting that the Ile-tRF-5X binding site mutations collaterally affected the inhibitory elements of the 3'UTR (**Fig 10B**). Addition of Ile-tRF-5X was still able to restore Rluc activity, but at a reduced intensity and without a dose-response effect, suggesting a role for Ile-tRF-5X binding sites in mediating its gene regulatory effects.

Expression of the Rluc reporter gene, placed under the control of the MAP3K4 mRNA 3'UTR, was reduced by only 10% in miRNA-depleted, Dicer -/- HCT116 cells, compared to 60% in wild-type HCT116 cells, which is compatible with the loss of miRNA-induced downregulation through the MAP3K4 mRNA 3'UTR. In cells depleted of miRNAs, addition of Ile-tRF-5X alleviated Rluc reporter gene expression by a marginal, but statistically significant, 15% at the highest concentration (**Fig 10C**). These small gene regulatory effects of Ile-tRF-5X were lost upon mutation of the Ile-tRF-5X binding sites (**Fig 10D**).

Taken together, these results suggest that bacterial Ile-tRF-5X may modulate the gene downregulatory properties of human MAP3K4 3'UTR, which may occur through competition with Dicer-derived miRNAs. Ile-tRF-5X does not seem to exert gene downregulatory effects through the MAP3K4 mRNA 3'UTR.

## HCT116 WT or Dicer -/- cells + Rluc-MAP3K4 3'UTR WT or Mut.

**Fig 10. Bacterial Ile-tRF-5X counteracts the miRNA-induced gene downregulatory properties of MAP3K4 3'UTR.** (**A-D**) Human HCT116 cells either wild-type (WT; **A-B**) or Dicer -/- (**C-D**) were co-transfected with synthetic Ile-tRF-5X (0, 50, 100 nM) and a psiCHECK-II reporter construct (50 ng; see S7 Fig) in which the Rluc reporter gene was coupled with WT (A, C) or mutated (Mut., B, D) MAP3K4 3'UTR. The details of the applied mutations are indicated in the S1 Information. An unrelated 13-nt sequence (100 nM) was used as control (0 nM Ile-tRF-5X). Dual luciferase reporter gene activity assays were performed as described previously [54], with Fluc as a normalization reporter control. **Statistical analysis**. Data were calculated from three biological replicate measurements (n = 3; mean ± SD). One-way analysis of variance (ANOVA) and Holm-Šídák's multiple comparisons (post-hoc test) were used for statistical analysis. Statistically significant differences are indicated as follows: $^*$ p < 0.05; $^{**}$ p < 0.01; $^{***}$ p < 0.001. ns, not significant.

### Bacterial OMVs and Ile-tRF-5X dose-dependently enhance human cell proliferation

Since the mitogen-activated protein kinase (MAPK) pathways are known to connect extracellular signals to fundamental intracellular processes [58], we investigated if the upregulation of MAP3K4 triggered, directly or indirectly, cell proliferation, inflammation or apoptosis.

The effect of bacterial OMVs and Ile-tRF-5X on proliferation of HCT116 cells was determined using a colorimetric assay. Cell proliferation was moderately increased upon exposure to bacterial OMVs (**Fig 11A**), but more substantially in response to increasing concentrations of Ile-tRF-5X (**Fig 11B**).

The bacterial OMV-induced cell proliferation was not affected by pre-treatment of the bacterial OMVs with an antisense to Ile-tRF-5X (**S9 Fig**).

To identify the human cell MAPK signaling pathway potentially regulated by bacterial OMVs and Ile-tRF-5X, we measured transcript levels of p38α MAPK, the transcription factor c-Jun and the cell division cycle 25 A (Cdc25A) phosphatase. Both bacterial OMVs and

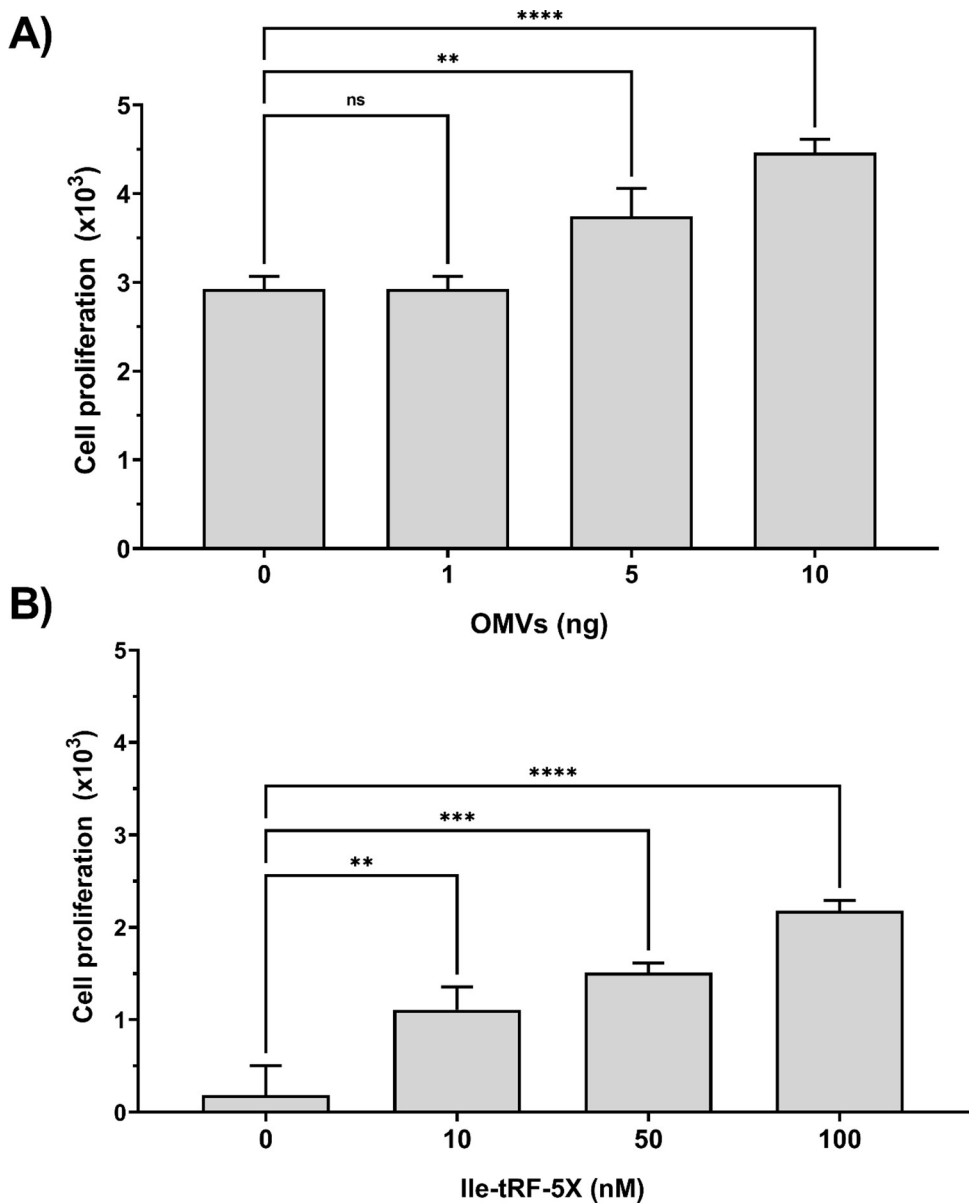

**Fig 11. Bacterial OMVs and Ile-tRF-5X dose-dependently enhance human cell proliferation.** (A-B) Human HCT116 cells were incubated with increasing amount of bacterial OMVs or transfected with increasing amount of Ile-tRF-5X. Cell proliferation was measured by plotting cell number from XTT-based absorbance (450 nm) measurements. Each dataset was normalized with its corresponding control (see materials and methods). **Statistical analysis**. Data were calculated from three biological replicate measurements (n = 3; mean ± SD), and each sample was tested in triplicate. One-way analysis of variance (ANOVA) and Holm-Šídák's multiple comparisons test (post-hoc test) was used for statistical analysis. Statistically significant differences (fold change vs. control) are indicated as follows: ** $p < 0.01$; *** $p < 0.001$; **** $p < 0.0001$. ns, not significant.

vsRNA Ile-tRF-5X promoted expression of Cdc25A and MAPK14 transcripts, whereas c-Jun mRNA seemed to be specifically modulated by OMVs, but not by Ile-tRF-5X (**S10 Fig**).

Finally, we performed a quantitative analysis of a dozen human cell transcripts relevant to inflammation and apoptosis to document their changes in expression levels in the presence of bacterial OMVs or Ile-tRF-5X (**S11 Fig**).

Some inflammatory cytokines, including interleukin-1 beta (IL-1β) and interleukin-6 (IL-6) were significantly upregulated by bacterial OMVs. In comparison, Ile-tRF-5X did not induce significant up or down regulation of the tested cytokines. The prototypical human inflammatory caspase-1 and the proapoptotic Bax were found to be upregulated in the presence of bacterial OMVs, whereas Ile-tRF-5X stimulated the survival pathway through upregulation of the apoptosis suppressor gene Bcl-2.

Taken together, these results suggest that bacterial OMVs and Ile-tRF-5X are potentially involved in the controlled regulation of human cell proliferation and death.

## Discussion

In this study, we have pursued our previous finding, in *E. coli* K-12 MG1655 and their OMVs, of an abundant 13-nt vsRNA [44], known as Ile-tRF-5X, and report on its potential role in bacteria-host interactions, and that of bacterial OMVs to act as a natural carrier. To simulate the intestinal microbiota environment, we chose the intestinal bacterium *E. coli*, which remains the most studied organism in modern biology [59], and the colonic epithelial HCT116 cell line as human host cells [60].

We observed that bacterial Ile-tRF-5X is differentially expressed in response to environmental factors. However, under normal laboratory conditions, either during the exponential or stationary phase, or upon inhibition of transcription and translation, the levels of Ile-tRF-5X in bacteria remained unchanged, although these processes are associated with a major shift in the abundance of tRNA [61, 62]. This uncoupling between tRF and tRNA levels in some conditions may seems intriguing but reinforces the idea that tRFs may have specialized roles different from their tRNA precursors. A lack of correlation has previously been reported between the accumulation of tRFs and different stresses applied to *Streptomyces coelicolor*, such as amino acid starvation, stringent response, or ribosome inhibition [63].

Stress conditions can lower the fraction of RNA polymerases that transcribe tRNA by about 60%, followed by degradation of the majority of bacterial tRNAs in less than twenty minutes [61, 64]. tRNA fragmentation is a highly conserved mechanism of the cellular response to stress in eukaryotes, as well [65]. Conversely, stress periods, including elevation of temperature [66], can also be a period of strong activation of the protein synthesis machinery [67]. The increase in bacterial Ile-tRF-5X levels might be explained not only by the controlled degradation of tRNAs (in this case, isoleucine) but also by their selective on-demand production by mechanisms that have yet to be fully understood. This duality in tRNA regulation, which may explain our results, at least in part, may be further supported by the demand-based model of tRNA degradation, developed by Sørensen and colleagues [68], suggesting that the tRNA pool is more dynamically regulated than previously thought. Moreover, previous findings have suggested that, in many respects, the metabolism of stable (tRNA and rRNA) and unstable (mRNA) RNAs are very similar [69] and, therefore, dynamically regulated to fit the bacteria's needs.

The increase of Ile-tRF-5X during environmental stress situations (closer to what could be encountered in real conditions) suggests potential regulatory functions. In eukaryotes, these functions are now established and continue to expand [70]. Recent studies in bacteria have provided rational evidence to support these hypotheses (reviewed in [25]). Lalaouna et al. [23] reported the existence of an *E. coli* tRF derived from an external transcribed spacer (ETS) capable of modulating the activities of the sRNAs RyhB and RybB, which coordinate the bacterial responses to iron deprivation [71] and cell envelope homeostasis [72], respectively. Furthermore, they suggest functional conservation of these fragments in the ETS and ITS (internal transcribed spacer) of tRNAs from many species belonging to Enterobacteriaceae family and an important major role for RNAse E in their release [23].

By transiently inhibiting RNAse E or RNase P, we found that only RNAse E, which play a central role in stable-RNA processing including two-thirds of pre-tRNAs [73], may contribute to Ile-tRF-5X levels. Initiation of tRNA maturation by RNase E is essential for cell viability in *E. coli* [74], and Ile-tRF-5X is formed of the first 13 nucleotides of the mature tRNA-Ile. RNase E is also known to be involved in the glycolytic pathway of the bacteria [75] and hence we can speculate that its modulation may vary depending on the culture condition (LB vs. M63) more specifically on the parameter of carbon source availability in the M63 medium. The accumulation of tRF observed in M63 could therefore be, among other effects, the result of a decrease in RNase E efficiency due to its nutritional dependence. In fact, the role of RNase E in tRNA maturation is well known [49, 74, 76] but until now its potential contribution to tRF generation has never been documented and our study provides a new insight in this direction.

The lack of RNase P involvement may be explained by recent findings stipulating that the essential function of RNase P is not the 5' maturation of pre-tRNAs, but rather the generation of pre-tRNAs from polycistronic operons [77].

Unlike tRNA genes that belong to tRNA operons [78], the tRNA-Ile gene is surrounded by rRNAs (*rrnA*, *rrnD*, and *rrnH*) operons, in a region with high evolutionary conservation which even serves as a tool for species characterization [79, 80]. More specifically, tRNA-Ile is part of the 16S-23S rRNA ITS [81]. It is worth recalling that tRNA and tRF processing may involve multiple independent pathways and multiple ribonucleases with overlapping functions [48]. This complexity made it challenging to identify the key enzymes mediating Ile-tRF-5X biosynthesis and/or turnover. The 16S and 23S rRNAs are respective components of the small and large ribosomal subunits that ultimately form the functional 70S ribosome [82]. The coexistence of tRNAs in ITS of rRNA operon remains enigmatic although widely reported [83]. It has been shown that all seven *E. coli* rRNA operons (including the 3 hosting the Ile-tRNA gene) are needed for optimal adaptation to changing physiological conditions [84]. Such localization of tRNA genes (and, therefore, tRFs) is probably not a coincidence, and may confer a strategic, functional advantage. Ideally, we would have needed a mutant strain of the isoleucine tRNA and another of the Ile-tRF-5X to better endorse the precise contribution at either the intracellular level or in OMVs. However, given the location of the genes, the removal of the three source sites would have led to adverse effects that would have biased data interpretation.

The selective enrichment of vsRNAs in OMVs and their effective transfer to human host cells were previously evidenced by other teams [34, 40, 85, 86]. The various OMV entry routes into host cells are discussed by O'Donoghue and Krachler [87], but the pathway(s) leading to their secretion from the parent bacterial cell is not well understood. Nevertheless, our study provides a different angle of perceiving these interactions in the largest symbiotic ecosystem in the gut microbiota [88], although it can hardly represent such a complex system and the biological events that take place.

Bacterial vsRNAs secreted in OMVs can potentially target a wide range of human host transcripts, which they may modulate to the advantage of the bacteria or, in a "commensal perspective", control a win-win balance. This phenomenon implies the existence of an endless number of highly regulated exchange/pathways between bacteria and the host cells. We focused on host MAP3K4, whose family (MAPKs) had already been reported as potentially involved in the cascade of reactions triggered by bacteria and/or their tRFs [39, 40, 70, 89], besides having identified dozens of binding sites for Ile-tRF-5X in this mRNA transcript.

The quantification of Ile-tRF-5X, either post-transfection or post-incubation with bacterial OMVs, suggests that Ile-tRF-5X remains stable in human cells, which is probably conferred by its secondary hairpin structure, in addition to the contribution of their parental tRNA biotypes. This apparent stability might also be conferred by binding to effector protein complexes, as it is the case for some eukaryotic tRFs [90]. The Ile-tRF-5X transfection and OMV

incubation approaches that we have used with human host cells have their own limitations, although comparable amounts of Ile-tRF-5X were used in this assay. Nevertheless, we noticed that the Ile-tRF-5X delivered by bacterial OMVs had a slightly higher rate of decrease than those directly transfected, suggesting a potential protective effect of OMVs or a slower internalization and/or intracellular release. However, the differences in the delivery for OMVs-associated and synthetic Ile-tRF-5X could affect these observations.

We observed that the MAP3K4 target candidate is upregulated at the mRNA level when human host cells are treated with bacterial OMVs, their sRNA content or Ile-tRF-5X alone. Bacterial RNA is a potent trigger for cell innate immune activation [91] and, in this respect, there is a high potential for MAPK pathways to be activated. This is in accordance with Tsatsaronis et al. [39], who proposed that the RNA content of extracellular vesicles may be sensed by endosomal receptors and, through a series of cascade reactions, lead to the activation of MAPKs.

Only bacterial OMVs induced the upregulation of MAP3K4 at the protein level. The decoupling of protein from mRNA levels may be explained, apart their measurement at a single time point, by the complexity of MAP3K4 transcript post-transcriptional modifications [92]. Being sensitive to a myriad of extracellular stimuli, MAPKs have developed a signal specificity related to the duration and magnitude of their interaction activities [93]. Here, OMVs appear to provide additional signal enhancement elements that vsRNAs do not have.

The binding of miRNAs involves the seed sequence, which is, on average, only 6–8 nt long, largely below the length of the vsRNA Ile-tRF-5X. From a molecular basis, our results suggest a model where Ile-tRF-5X and other vsRNAs compete with the pool of miRNAs regulating the 3'UTR of MAP3K4. By limiting the (down)regulatory effects of miRNAs, Ile-tRF-5X vsRNAs would contribute to the stabilization of MAP3K4 transcripts. The absence of Dicer and of most miRNAs, which are known to occupy binding sites in mRNA 3'UTRs, would be expected to facilitate access of Ile-tRF-5X to the 3'UTR of MAP3K4 mRNA. Under these conditions, Ile-tRF-5X does not seem to have intrinsic gene regulatory properties and may not be a simple on/off switch, but rather a modulator with sophisticated effects on phenotype [94].

This is what we have also reported in a previous study where we demonstrated that a 12-nt miRNA fragment (a "semi-miRNA"), without evidence of any involvement in a silencing complex, was able to modulate the activity of the miRNA from which it derived [95]. Similar to Ile-tRF-5X, semi-miRNA were probably too short to have direct gene regulatory properties [95]. It may not be excluded that miRNAs from non-canonical biogenesis pathways [96] could take over to interact differently with vsRNAs, such as Ile-tRF-5X. Of over fifteen miRNAs identified as potential MAP3K4 inhibitors, only miR-148a, a member of the miR-148/152 family, has been validated by three strong methodologies (reporter assay, qPCR, Western blot) [55, 97]. The seed sites of miR-148a do not overlap with those of Ile-tRF-5X but are 40 nt apart.

To study the global role of miRNAs, Dicer knockout models are often used. However, it is important to point out that Dicer is not only involved in the maturation of miRNAs and is probably implicated in several other regulatory processes [98], thus making the "Dicer phenotype" more challenging to interpret and hardly solely attributable to miRNA depletion.

It is difficult to assess the effect of vsRNAs on the entire MAP3K4 transcript structure and, more specifically, on its 3'UTR. However, beyond on-site competition, it is possible that their interactions may cumulatively or cooperatively lead to structural modifications that contribute to the post-transcriptional modulation of MAP3K4. Moreover, the possible cooperative effect of miRNAs [99] on the 3'UTRs may be perturbed by vsRNAs, resulting in an alleviation of the repression or, more drastically, a loss of miRNA function.

Bacterial Ile-tRF-5X has also predicted binding sites in the 5'UTR of MAP3K4 mRNA, a region where interactions can often lead to activation of transcription [100], which may

contribute to increase MAP3K4 mRNA levels. Despite the normalization, which accounts for the variation in transfection efficiency and cell viability [101], the dual luciferase assay system made use of the MAP3K4 mRNA 3'UTR and, therefore, does not take into account the possible contribution of the ORF or 5'UTR, or the expression level of the endogenous target (here, MAP3K4). The list of potential factors involved in the modulation of MAP3K4 via vsRNAs is therefore non-exhaustive.

MAP3K4 is a ubiquitously expressed component of the stress-activated MAPK signaling module, which directly phosphorylates and activates the JNK and p38 families of MAPKs [102]. Depending on the context, MAP3K4 is either a proto-oncogene or a tumor suppressor. MAP3K4 stimulation may result in cell proliferation, a cellular response downstream of the signaling cascade regulated by MAP3Ks [58]. Although presenting several redundancies and crossovers, the two main MAPK signaling pathways (p38 and JNK) are differentially stimulated by bacterial OMVs and Ile-tRF-5X. OMVs seem to be able to engage indiscriminately both p38 and JNK pathways, while Ile-tRF-5X seems to be more specific to the p38 pathway (**S10 Fig**). Furthermore, p38 is required for cell proliferation and survival in colorectal cancer cells [103]. MAPKs constitute a dense network of pathways (crosstalk) that can hardly be partitioned from the functional point of view [104].

Several studies have clearly established these links between (endogenous) tRF or MAPKs expression and cell proliferation [19, 105, 106]. Host cell proliferation following incubation with OMVs has also been described in the literature [107, 108]. The colorimetric assay is also a mirror of viability and toxicity of metabolically active cells. Thus, fluctuation in proliferation may be related to mitochondrial abundance and cell viability (vs. death). Given the discrepancy observed in the induction of Bax (pro-death) and Bcl-2 (pro-survival) transcripts by OMVs, on one hand, and the vsRNA Ile-tRF-5X on the other hand, we can speculate on an in-balance bacterial modulation between cell proliferation and host cell death program. The specific induction of Caspase-1 by bacterial OMVs may trigger pyroptosis instead of apoptosis, as documented in previous studies [109]. It is necessary, however, to see at the protein level if all these patterns and phenotypes are confirmed and more precisely how they are affected in an in vivo experiment.

Finally, it has been shown that RNA fragments from plants, bacteria and fungi are ubiquitous in human plasma [110, 111] and may be delivered to any organ. Considering the present study in this broader perspective, bacterial OMVs and sRNAs, such as Ile-tRF-5X, may serve as universal mediators of microbial communication [39] and biomarkers of health status [111]. Also, on the host side, miRNAs were found to directly affect the growth of gut bacteria and were deemed essential for their maintenance [112]. The development of HTS technologies and the growing interest in bacterial OMVs and their RNA content being transferred to human host cells provide an unprecedented opportunity to rebuild our current understanding of host-bacteria relationships, which may either be pathogenic or symbiotic. Our study pleads in favor of studying vsRNAs, like the 13-nt Ile-tRF-5X, to provide new perspectives on the existing symbiosis between bacteria and their hosts. As an illustrative overview, we propose a schematic diagram representing how bacterial OMVs and their RNA content may be involved in the preservation of the delicate, dynamic and complex balance of homeostasis (**Fig 12**).

## Materials and methods

Critical materials and resources are summarize in Table 1.

### Bacterial strains and culture conditions

The bacterial strains used in this study and their experimental conditions are summarized in **S1 Table**. Unless otherwise stated, the reference strain of the study is *E. coli* MG1655. Bacteria

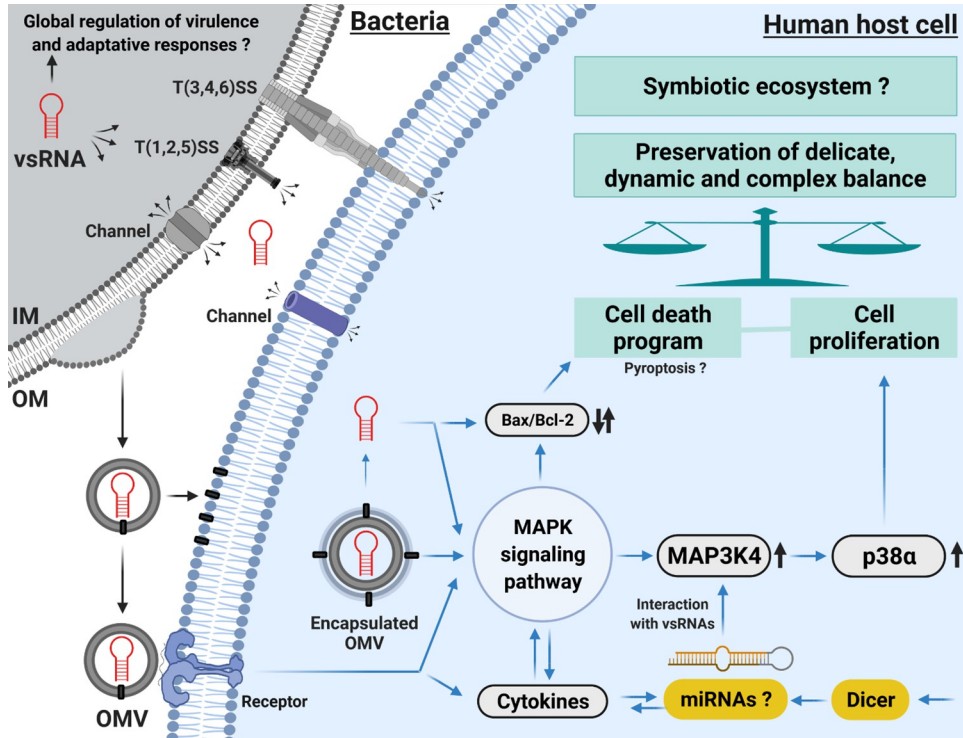

**Fig 12. Schematic representation of how bacterial OMVs and their content in vsRNAs, such as Ile-tRF-5X, may be involved in preserving the delicate, dynamic and complex balance of homeostasis.** IM = Inner Membrane; OM = Outer Membrane; OMV = Outer Membrane Vesicle; T(1,2,3,4,5,6)SS = Type (1,2,3,4,5,6) Secretion System. Original figure created with BioRender.com based on our previous review [27]. Not drawn to scale.

were grown in Luria-Bertani (LB) medium or in M63 Glucose with 250 RPM aeration and on LB agar at 37˚C.

## Human cell line culture

The human colorectal carcinoma cell lines HCT116 Wild type and Dicer-/- were grown in complete Dulbecco's modified Eagle's medium (DMEM, Wisent, St-Bruno, Canada, cat. no. 219-010-XK) supplemented with 10% fetal bovine serum, 1 mM L-glutamine, 100 units/ml penicillin, and 100 µg/ml streptomycin. Cells were grown and maintained in tissue culture plates and incubated at 37˚C in a humidified atmosphere under 5% $CO_2$. Cells were kept in the exponential growth phase and subcultured every 2–3 days.

## Plasmid constructs

The complete 3'UTR of MAP3K4 (4991–5506 nt, NCBI accession number NM_005922.4) was amplified by PCR and introduced downstream of the *Renilla luciferase* (Rluc) reporter gene in the XhoI/NotI cloning sites of the psiCHECK-II vector (Promega) to generate psiCHECK-II-3UTR-MAP3K4-WT.

A reporter construct bearing a mutated version of the 3'UTR of MAP3K4 was also engineered by using gBlocks gene fragments (Integrated DNA Technologies, Inc.) introduced in the XhoI/NotI cloning sites of psiCHECK-II vector to produce psiCHECK-II-3UTR--MAP3K4-MUT. The mutations (transversion substitution) were designed to abolish the interactions with the vsRNA Ile-tRF-5X and affect every other nucleotide of the binding sites.

**Table 1. STAR★ Methods—Key Resources.**

| REAGENT or RESOURCE | SOURCE | IDENTIFIER |
|---|---|---|
| Bacterial Strains | | |
| *Escherichia coli* MG1655 WT (EM1055) | Dr. Eric Massé | See S1 Table |
| *Escherichia coli* rne-3071 ts (EM1277) | | |
| *Escherichia coli* rnpA ts (KP1036) | | |
| Chemicals, Peptides, and Recombinant Proteins | | |
| Polyethylenimine | Sigma | Cat# 919012 |
| Lipofectamine 2000 | Invitrogen | Cat# 11668019 |
| Opti-MEM | Invitrogen | Cat# 31985062 |
| PKH67 dye | Sigma | Cat# MINI67 |
| HBSS 1X | Wisent | Cat# 311-513-CL |
| CellTracker Red CMTPX Dye | Invitrogen | Cat# C34552 |
| SlowFade Gold Antifade Mountant | Invitrogen | Cat# S36942 |
| TRIzol | Invitrogen | Cat# 15596026 |
| RNAzol RT | Sigma | Cat# R4533 |
| SSo Advanced SYBR Green mix | Biorad | Cat# 1725271 |
| RIPA buffer | Cold Spring Harbor | CSH protocols |
| Complete EDTA-free Protease Inhibitor cocktails | Roche | Cat# 4693132001 |
| PhosSTOP | Roche | Cat# 4906845001 |
| anti-MAP3K4 (mouse monoclonal) | SCBT | Cat# sc-166197 |
| anti-β-tubulin (mouse monoclonal) | SCBT | Cat# sc-5274 |
| Critical Commercial Assays | | |
| Dual-Luciferase Reporter Assay kit | Promega | Cat# E1980 |
| Cell Proliferation Kit II | Roche | Cat# 11465015001 |
| Exobacteria Kit | S. Biosciences | Cat# EXOBAC100A-1 |
| VacuCap filters | Pall | Cat# 4634 / Cat# TA4632 |
| RiboPure Bacteria Kit | Invitrogen | Cat# AM1925 |
| HiFlex miScript II RT Kit | Qiagen | Cat# 218160 |
| MicroAmp Fast Optical 96-Well Plate | Applied Biosystem | Cat# 4346907 |
| miRCURY LNA RT Kit | Qiagen | Cat# 339340 |
| miRCURY LNA SYBR Green PCR Kit | Qiagen | Cat# 339346 |
| Multiplate PCR plate | Biorad | Cat# MLL9601 |
| Clarity Max Western ECL Substrate | Biorad | Cat# 1705062 |
| Deposited Data | | |
| Raw and processed data from sRNA-Seq analysis | Ref [44] | BioProject accession: PRJNA826503 GEO accession: GSE200758 DOI: https://www.ncbi.nlm.nih.gov/geo/query/acc.cgi?acc=GSE200758 |
| Experimental Models: Cell Lines | | |
| HCT116 WT | ATCC | Cat# ATCC CCL-247 |
| HCT116 Dicer -/- | This paper | NA |
| Oligonucleotides | | |
| Primers | IDT | S2 Table |
| Recombinant DNA | | |
| psiCHECK-II vector | Promega | Cat# 8021 |
| psiCHECK-II + MAP3K4 3'UTR WT | This paper | S1 Information |
| psiCHECK-II + MAP3K4 3'UTR MUT | This paper | S1 Information |

*(Continued)*

**Table 1.** (Continued)

| REAGENT or RESOURCE | SOURCE | IDENTIFIER |
|---|---|---|
| Software and Algorithms | | |
| Volocity 4.2.1 software | Quorum Technologies | Technical resources/software/downloads/ |
| Primer-BLAST | Ref [118] | Primer-BLAST website |
| StepOne Software | Thermofisher | Technical resources/software/downloads/ |
| Image Studio Lite | Image Studio | Image studio lite website |
| ImageJ | Ref [125] | Technical resources/software/downloads/ |
| GraphPad Prism 9.2.0 | GraphPad | Technical resources/software/downloads/ |
| Other | | |
| Luminometer | TECAN | Cat# Infinite M1000 PRO |
| Sorvall WX+ Ultracentrifuge (T-1250 Fixed Angle Rotor) | Thermo Scientific | Cat#75000100 |
| Beckman Coulter Optima Ultracentrifuge (TLA 100.4 rotor) | Beckman | Cat# TL 100 |
| Wave FX-Borealis—Leica DMI 6000B | Quorum Technologies | NA |
| ImagEM camera | Hamamatsu | NA |
| NanoDrop 2000 Spectrophotometer | Thermo Scientific | Cat# ND-2000 |
| StepOne Real-Time PCR System | Thermofisher | Cat# 4376357 |
| CFX Connect Real-Time PCR Detection System | Biorad | Cat# 1855200 |
| C-DiGit Blot Scanner | LI-COR Biosciences | Cat# 3600 |

These constructions are described in **S1 Information.** All the constructs were confirmed by DNA sequencing at the Plateforme de Séquençage et Génotypage des Génomes (Centre de Recherche du CHU de Québec–CHUL, QC, Canada).

## Cell transfection and dual luciferase assay

Three hundred thousand cells were cultured in 6-well plates and transfected the following day at 70–80% confluency using polyethylenimine [113] (PEI; Sigma, ON, Canada, cat. no. 919012) or lipofectamine 2000 [114] (Invitrogen, ON, Canada, cat. no. 11668019) as described previously with slight modifications. The vsRNAs or plasmids to be transfected as well as the transfection reagents were diluted in Opti-MEM (Invitrogen, ON, Canada, cat. no. 31985062).

Forty-eight [48] h after transfection with psiCHECK-II vectors, cells were washed with PBS and lysed with 500 µl of passive lysis buffer. Luciferase activities were measured using the Dual-Luciferase Reporter Assay System (Promega, WI, USA, cat. no. E1980) on a luminometer (TECAN INFINITE M1000 PRO), according to the manufacturer's instructions. Rluc expression was reported relative to the expression of the internal *Firefly luciferase* (Fluc) reporter control. Rluc expression was further normalized to the control in which cells were co-transfected with a synthetic mock vsRNA, referred to as mock control. All assays were conducted in triplicates in 96-well plates.

## Cell proliferation assay

For the quantification of cellular proliferation, we used the Cell Proliferation Kit II (Roche, Mannheim, Germany, cat. no. 11465015001), a colorimetric assay based on XTT (sodium 3

′-[1- (phenylaminocarbonyl)- 3,4- tetrazolium]-bis (4-methoxy6-nitro) benzene sulfonic acid hydrate). A standard range of 0–30,000 cells were used to define the appropriate cell concentration and incubation time for the cell proliferation test (S1 Fig). Next, HCT116 cells were incubated in the presence of bacterial OMVs or transfected with vsRNAs at different concentrations in 96-well plates. After 48 h, the XTT labeling mixture was added to the microplate. After 3 h of incubation, the absorbance was measured using a microplate (ELISA) reader (spectrophotometer) at 450 nm wavelength.

## Isolation and characterization of OMVs

Outer membrane vesicles (OMVs) were purified from the reference strain *E. coli* K-12 MG1655, as described previously with some modifications [115, 116] or with the Exobacteria Kit (System Biosciences, CA, USA, cat. no. EXOBAC100A-1) following the manufacturer's instructions.

Briefly, *E. coli* K-12 MG1655 was grown in 120 ml of LB to an $OD_{600}$ of ~0.5, after which the supernatant was collected by centrifugation at 10,000 x g for 10 min at 4˚C. The supernatant was filtered through a 0.45 and 0.2 μm-pore size VacuCap filters (PALL, MI, USA, cat. no. 4634, TA4632), after which an inoculum was taken to confirm the absence of bacteria on LB agar. Bacterial OMVs were then washed twice with PBS and pelleted each time by ultracentrifugation at ~200,000 x g for 2 h at 4˚C in a Thermo Scientific Sorvall WX+ Ultracentrifuge with the T-1250 Fixed Angle Rotor. After removing the supernatant, OMVs were resuspended in 200 μl sterile PBS.

Isolation of OMVs by the Exobacteria Kit use a precipitation-free gravity column system. Based on the same approach, 30 ml of pre-cleared (centrifuged and filtered as above) bacterial culture media were used to harvest OMVs following the manufacturer's instructions. Bacterial OMVs were collected in 1.5 ml of elution buffer.

Purified OMVs were subjected to quality control including agar plating to ensure lack of bacterial contamination, Coomassie blue stained gel, transmission electron micrograph (TEM) and dynamic light scattering (Zetasizer). These steps have been described in detail elsewhere [44]. Purified OMVs were stored at -80˚C before being used for downstream applications.

## Confocal microscopy

Bacterial OMVs were stained with the green, fluorescent membrane dye PKH67 (Sigma, ON, Canada, cat. no. MINI67) at 1 μM final concentration in Diluent C (Sigma) for 5 min. The stained OMVs were then pelleted at 30,000g for 1 h in a TL-100 Optima Ultracentrifuge (Beckman) using rotor TLA 100.4. PKH67-labeled OMVs were washed in HBSS 1X (Wisent, QC, Canada, cat. no. 311-513-CL) to get rid of the free dye and then incubated with the fluorescently labeled HCT116 cells. HCT116 cells were labeled with the CellTracker Red CMTPX Dye (Invitrogen, ON, Canada, cat. no. C34552) for 30 min at 37˚C in darkness and washed with PBS.

After incubation for the indicated times, trypsin was used for cell dissociation followed by centrifugation at room temperature for 5 min. Cells were then resuspended in 100 μl 2% PFA and kept for 15 min at room temperature to allow fixation. Cell deposition on slides was performed using cytospin. One drop of Gold Antifade Mountant with DAPI (SlowFade, ON, Canada, cat. no. S36942) was applied directly to the slide, which was then sealed with varnish and imaged using a Wave FX-Borealis—Leica DMI 6000B (Quorum Technologies) microscope (63X) with ImagEM camera (Hamamatsu, 512x512 pixels) at the CHUL–Université Laval Bio-

imaging platform. The images were processed with the Volocity 4.2.1 software (Quorum Technologies).

## RNA isolation

Total RNA from bacterial strains was extracted using the hot phenol procedure [117] or the RiboPure Bacteria Kit (Invitrogen, ON, Canada, cat. no. AM1925) following the manufacturer's recommendations. Total RNA from the human cell lines was extracted using TRIzol (Invitrogen, ON, Canada, cat. no. 15596026) or RNAzol RT (Sigma, MO, USA, cat. no. R4533) reagents following the manufacturer's recommendations. Bacterial OMV RNA was isolated with RNAzol RT (Sigma, MO, USA, cat. no. R4533) reagents following the manufacturer's recommendations (allow selective isolation, enrichment, of small RNAs <200 nt). All RNA samples were subjected to treatment with DNase I when applicable, quantified with the NanoDrop 2000 Spectrophotometer (Thermo Scientific, cat. no. ND-2000) and kept at -80°C for subsequent experiments.

## RT-qPCR

cDNA was generated by reverse transcription using the HiFlex miScript II RT Kit (Qiagen, MD, USA, cat. no. 218160) from 1 μg of DNase-treated RNA following the manufacturer's protocol. After diluting the cDNA (1/10), qPCR was performed using the SSo Advanced SYBR Green mix (Bio-Rad, CA, USA, cat. no. 1725271) in 0.1 ml MicroAmp Fast Optical 96-Well Reaction Plate (Applied Biosystem, cat. no. 4346907).

The final concentration of the primers (Integrated DNA Technologies, Inc.) used in RT-qPCR was 500 nM and their sequence are listed in **S2 Table**. The primers were designed with Primer-BLAST tools and recommendations [118]. Primers were chosen to allow specific amplification of the target messengers (span exon-exon junction). Temperature gradient tests were performed to determine the best annealing temperature for each primer pair.

Unless otherwise specified, qPCR reactions were performed using the StepOne Real-Time PCR System (cat. no. 4376357). Unless otherwise specified, all data obtained (from StepOne Software) were normalized with reference genes (ACTB or GAPDH) and reported to the controls. The relative quantitation was calculated using the ΔΔCt method [119].

## vsRNA detection by splint-ligation-based strategy

The splint-ligation detection of vsRNA Ile-tRF-5X, which also relies on data from the literature on small RNA quantification [120, 121], was described previously [122] and is illustrated in **S12 Fig**. For the highest efficiency and specificity, several adjustments were carried out at the pre-annealing (splint+adapter+vsRNA), ligation, reverse transcription, and qPCR steps. We employed a ligation step (T4 RNA ligase) to elongate vsRNA Ile-tRF-5X prior to RT-qPCR. A 13-nt adapter was tagged to the sequence of interest (Ile-tRF-5X) with the use of a specific 20-nt splint.

The adapter-ligated vsRNA and total RNA content were converted to cDNA with the miR-CURY LNA RT Kit (Qiagen, MD, USA, cat. no. ID 339340). Specific and sensitive custom miRNA primer set designed by the Exiqon experts and optimized with LNA technology [123] were used.

After diluting the cDNA (1/10), qPCR was performed using the miRCURY LNA SYBR Green PCR Kit (Qiagen, MD, USA, cat. no. 339346) with CFX Connect Real-Time PCR Detection System (Bio-Rad, cat. no. 1855200) in 96-well plates (Multiplate PCR plate, cat. no. MLL9601) following the manufacturer's protocol. The UniSp6 RNA spike-in was used for

cDNA synthesis and PCR amplification normalization. In bacteria, 23S and/or 16S served as reference genes.

## Protein extraction

Unless otherwise specified and in order to obtain RNA and protein from the same sample, proteins were extracted directly from the organic part remaining after RNA extraction with TRIzol reagent (Invitrogen, cat. no. 15596026). A sonication step was included to ensure better protein dissolution (5 cycles of 15 sec sonication and 30 sec ice incubation; sonicator settings: amplitude 80%, pulses 90%).

Homemade radioimmunoprecipitation assay (RIPA) buffer [124] was also used to extract OMV-associated proteins. One tablet of each of the complete EDTA-free Protease Inhibitor cocktails (Roche, QC, Canada, cat. no. 4693132001) and PhosSTOP (Roche, QC, Canada, cat. no. 4906845001) was added per 10 ml 1X RIPA buffer.

## Western blot

Protein extracts were analyzed by 10% (wt/vol) SDS-PAGE and immunoblotting using anti-MAP3K4 (Mitogen-activated protein kinase kinase kinase 4, SCBT, TX, USA, cat. no. sc-166197), anti-β-tubulin (SCBT, TX, USA, cat. no. sc-5274) antibodies. Chemiluminescence detection was performed using C-DiGit Blot Scanner (LI-COR Biosciences) with Clarity Max Western ECL Substrate (Bio-Rad, CA, USA, cat. no. 1705062). The images were acquired with Software Image Studio Lite (Image Studio) and densitometry analyses were performed with ImageJ [125].

## Statistical analysis

The statistical method used is mentioned in each figure legend. All statistical analyses were performed using GraphPad Prism version 9.2.0 (GraphPad Software, Inc., La Jolla, CA, USA), with statistical significance set at $p < 0.05$. Details of the tests are mentioned under each figure when applicable.

## Contact for reagent and resource sharing

Further information and requests for resources and reagents should be directed to and will be fulfilled by the Lead Contact, Patrick Provost (patrick.provost@crchudequebec.ulaval.ca).

## Materials availability

This study did not generate new unique reagents.

## Supporting information

**S1 Fig. Setting up of the XTT test of cell proliferation.** Determination of the optimal measurement time and the appropriate amount of cells.
(DOCX)

**S2 Fig. Bacterial Ile-tRF-5X and mature Ile-tRNA levels under different experimental conditions.** (**A**) *E. coli* MG1655 bacteria were grown at 37˚C up to the exponential (Reference, R) and stationary phases of growth in either complete (rich) LB or minimal M63 medium. (**B**) *E. coli* MG1655 bacteria were grown in LB medium at exponential phase (R) in different temperatures: 30, 37 and or 44˚C. (**C**) Bacterial mRNA or protein synthesis was inhibited by addition of chloramphenicol or rifampicin, respectively, to cultures of *E. coli* MG1655 grown at

exponential phase (R). The level of bacterial Ile-tRF-5X level is not modulated by transcription or translation activity (**D**) *E. coli* strains carrying heat-sensitive (hs) mutations in the essential genes rne-3071-hs (EM1277) and rnpA-hs (KP1036) were grown in LB medium at 30˚C and then heat-shocked (44˚C) to transiently inhibit ribonuclease (RNase) P or RNase E, which are involved in tRNA maturation. Bacteria RNase E contributes to Ile-tRF-5X biogenesis. For more details, see **Supplementary S1 Table**. In all conditions, the level of Ile-tRF-5X and Ile-tRNA were measured by LNA RT-qPCR. A spike-in (UniSp6) and reference genes (23S and/or 16S) were used as control and for normalization. The results are reported in fold change compared to the reference condition. **Statistical analysis**. Data were calculated from three biological replicate measurements (n = 3; mean ± SD). Two-way analysis of variance (ANOVA) and Dunnett's multiple comparisons (fold change vs. reference test) or Šídák's multiple comparisons test (Ile-tRF-5X vs. Ile-tRNA) were used for statistical analysis. Statistically significant differences (fold change vs. reference or Ile-tRF-5X vs. Ile-tRNA shown in purple) are indicated by stars (*), * $p < 0.05$; ** $p < 0.01$; *** $p < 0.001$; **** $p < 0.0001$; ns, not significant. (DOCX)

**S3 Fig. Ile-tRF-5X level in different E. coli samples.** Ile-tRF-5X was quantified from normalized RNA-Seq data in *E. coli* exponential growth phase (reference, ref, R), stationary phase, and after treatment with chloramphenicol (R+cat), rifampicin (R+rif) or heat shock to transiently inactivate RNase E (RNase E -). Fig 2A (LB medium), Fig 3 and Fig 4 (RNase E) are the quantitative validation of these data. One biological replicate. See [reference [1]] for more details.
(DOCX)

**S4 Fig. Proportion in copy number of intracellular Ile-tRF-5X or OMV-packaged Ile-tRF-5X.**
(DOCX)

**S5 Fig. Confocal microscopy imaging of labelled OMVs taken up by human HCT116 cells.** HCT116 cells were stained with a Cell Tracker CMTPX™ (red) and then incubated for 2 h with bacterial OMVs labelled with PKH67 (green). The nuclei are stained with DAPI (blue). The XZ and YZ projection of Volocity shows that OMVs were internalized by HCT116 cells and localized mainly in the cytoplasm (See panel A at 2 h in Fig 5.).
(DOCX)

**S6 Fig. Human MAP3K4 mRNA harbors potential binding sites for bacterial Ile-tRF-5X.** Three bioinformatics tools (microRT, blastN and RNAhybrid) used in combination predict several Ile-tRF-5X (5X) binding sites in the human MAP3K4 mRNA. Base pairing of the top three Ile-tRF-5X binding sites is shown on the left. The number of nucleotides (mer) involved in the interaction (seed sequences), their minimal free energy (MFE, according to RNAhybrid) and their position (5'UTR, ORF or 3'UTR) are listed in the table. * indicates that the 13th nt of Ile-tRF-5X is a C, thus allowing perfect base pairing with MAP3K4 mRNA in its ORF.
(DOCX)

**S7 Fig. Schematic representation of the predicted Ile-tRF-5X and miRNA base pairing in the MAP3K4 mRNA 3'UTR.** The base pairing interactions between bacterial Ile-tRF-5X and human MAP3K4 mRNA (RNAhybrid) are shown in red, with the minimum free energy (Mfe) and position (nucleotides). The base pairing interactions between human miRNAs and MAP3K4 mRNA (Targetscan) are highlighted in grey. The nucleotides involved in the Watson–Crick base pairing are in bold. The numbers in square brackets ([]) correspond to the number of non-displayed nucleotides in the 3'UTR of MAP3K4 mRNA. The full MAP3K4

mRNA 3′ UTR sequence was cloned downstream of the humanized Rluc (hRluc) gene, in the dual-luciferase reporter gene expression vector psiCHECK-II, with hFluc as a normalization control. hRluc, humanized *Renilla luciferase* gene; hluc+, humanized *Firefly luciferase* gene.
(DOCX)

**S8 Fig. Secondary structure of 3'UTR MAP3K4 displaying the binding sites of miRNAs and Ile-tRF-5X.** FORNA Web-based tools were used to illustrate RNA secondary structure of the full 3'UTR of MAP3K4. In blue we have the binding site positions of miRNAs and in red those of Ile-tRF-5X. For each of these actors, the positions (P) of the nucleotides (nt) are indicated.
(DOCX)

**S9 Fig. OMVs enhance HCT116 cell proliferation.** Cell number was estimated from XTT-based absorbance (450 nm) measurement. The % of cell proliferation was deduced from that of the mock control set at 100%. The use of Ile-tRF-5X (as5X, 100nM) did not significantly reduce cell proliferation. Each data set is normalized with its corresponding control. **Statistical analysis**. All data presented were calculated from three biological replicate (n = 3) measurements ± SD and each sample was tested with 3 replicates. The one-way analysis of variance (ANOVA) and Holm-Šídák's multiple comparisons test were used for statistical analysis. Statistically significant differences (fold change vs. mock) are indicated as follows: $^{*}$ $p < 0.05$; $^{***}$ $p < 0.001$; $^{****}$ $p < 0.0001$.
(DOCX)

**S10 Fig. Relative quantification of MAPK14 (p38α), CDC25A and c-JUN expressions at mRNA level in HCT116 cells by RT-qPCR after incubation with OMVs or transfection with Ile-tRF-5X.** Data were normalized with a reference gene (ACTB), reported to mock control, and expressed with a relative quantitation method (ΔΔCt). **Statistical analysis**. All data presented were calculated from three biological replicate (n = 3) measurements ± SD. The one-way analysis of variance (ANOVA) and Dunnett's multiple comparisons were used for statistical analysis. Statistically significant differences (fold change vs. mock) are indicated by stars (*), $^{*}$ $p < 0.05$; $^{**}$ $p < 0.01$; $^{***}$ $p < 0.001$, $^{****}$ $p < 0.0001$.
(DOCX)

**S11 Fig. Changes in expression of cytokines and apoptosis factors in HCT116 cells after incubation with bacterial OMVs or transfection with Ile-tRF-5X.** Relative mRNA expression was quantified by RT-qPCR. Data were normalized with a reference gene (ACTB), reported as fold change vs mock control, and expressed with the relative quantitation method (ΔΔCt). **Statistical analysis**. Data were calculated from three biological replicate measurements (n = 3; mean ± SD), and each sample was tested in triplicate. Two-way analysis of variance (ANOVA) and Holm-Šídák's multiple comparisons test (post-hoc test) were used for statistical analysis. Statistically significant differences (fold change vs mock) are indicated as follows: $^{*}$ $p < 0.05$.
(DOCX)

**S12 Fig. vsRNA monitoring (Splint-ligation-based strategy).** A) Experimental scheme for the detection of vsRNAs, here Ile-tRF-5X as example (See mat. & meth. section). B) specificity test of Exiqon Ile-tRF-5X primers. The primers can discriminate Ile-tRF-5X from any other sequence by up to one nucleotide difference using LNA technology. C) Single peak melt curve assessing the specificity of amplification. Ile-tRF-5X melt curve analysis shows the production of specific and single product. D) comparative analysis of Ile-tRF-5X quantifications by RNA-seq (semi-quantitative) and by qPCR reported in percentage. Ile-tRF-5X was quantified in *E.*

*coli* exponential growth phase (reference = R), stationary phase, after treatment with rifampicin (R+rif) or chloramphenicol (R+cat), or finally in derived OMVs.
(DOCX)

**S1 Table. List of bacteria used in the study and their growth conditions.**
(DOCX)

**S2 Table. Primers used in the study.**
(DOCX)

**S1 Information. Details of the design of the plasmid construction for the dual luciferase assay with MAP3K4 3'UTR WT (A) and MAP3K4 3'UTR MUT (B).**
(DOCX)

## Acknowledgments

We thank Katheryn Ouellet-Boutin for technical assistance.

## Author Contributions

**Conceptualization:** Idrissa Diallo, David Lalaouna, Eric Massé, Patrick Provost.

**Data curation:** Idrissa Diallo, Jeffrey Ho.

**Formal analysis:** Idrissa Diallo, Jeffrey Ho, Marine Lambert, Abderrahim Benmoussa, Zeinab Husseini.

**Funding acquisition:** Patrick Provost.

**Investigation:** Idrissa Diallo, David Lalaouna.

**Methodology:** Idrissa Diallo, Marine Lambert, Abderrahim Benmoussa, Zeinab Husseini, David Lalaouna.

**Project administration:** Patrick Provost.

**Resources:** David Lalaouna, Eric Massé, Patrick Provost.

**Software:** Jeffrey Ho.

**Supervision:** Patrick Provost.

**Validation:** Idrissa Diallo, Jeffrey Ho, David Lalaouna, Patrick Provost.

**Visualization:** Idrissa Diallo.

**Writing – original draft:** Idrissa Diallo.

**Writing – review & editing:** Idrissa Diallo, Jeffrey Ho, Marine Lambert, Abderrahim Benmoussa, Zeinab Husseini, David Lalaouna, Eric Massé, Patrick Provost.

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
