## [Decision Letter · Decision Letter 0]

12 May 2022

Dear Dr. Provost,

Thank you very much for submitting your manuscript "A tRNA-derived fragment present in E. coli OMVs regulates host cell gene expression and proliferation" for consideration at PLOS Pathogens. As with all papers reviewed by the journal, your manuscript was reviewed by members of the editorial board and by several independent reviewers. In light of the reviews (below this email), we would like to invite the resubmission of a significantly-revised version that takes into account the reviewers' comments.

Both reviewers agreed that the work is interesting, well done and of general interest to the broader scientific community working on regulatory RNAs and host-pathogen interactions.

However, there are several important points that the authors have to address. Among them, the previous results must be accessible to reviewers AND readers. I recommend that the authors publish the submitted work (Diallo et al.,manuscript submitted) on a preprint repository such as bioRxiv. If this is not possible, authors should present the previous data in their rebuttal letter. However, I do not encourage the latter as this present work should not be published without the publication of the previous work.

We cannot make any decision about publication until we have seen the revised manuscript and your response to the reviewers' comments. Your revised manuscript is also likely to be sent to reviewers for further evaluation.

Sincerely,

Mathieu Coureuil

Guest Editor

PLOS Pathogens

Xavier Nassif

Section Editor

PLOS Pathogens

Kasturi Haldar

Editor-in-Chief

PLOS Pathogens

orcid.org/0000-0001-5065-158X

Michael Malim

Editor-in-Chief

PLOS Pathogens

orcid.org/0000-0002-7699-2064

Both reviewers agreed that the work is interesting, well done and of general interest to the broader scientific community working on regulatory RNAs and host-pathogen interactions.

However, there are several important points that the authors have to address. Among them, the previous results must be accessible to reviewers AND readers. I recommend that the authors publish the submitted work (Diallo et al.,

manuscript submitted) on a preprint repository such as bioRxiv. If this is not possible, authors should present the previous data in their rebuttal letter. However, I do not encourage the latter as this present work should not be published without the publication of the previous work.

Reviewer's Responses to Questions

**Part I - Summary**

Reviewer #1: In this manuscript, Diallo et al describe the regulatory impact of an abundant short RNA from outer membrane vesicles (OMVs) of Escherichia coli, isoleucine tRNA-derived fragment Ile-tRNA-5X, on the components of mitogen-activated protein kinase pathway in human host cells.

The authors emphasize an important point on the RNA-based crosstalk during host-pathogen interactions and the role of tRNA-fragments in these cross-regulations.

They monitored the amount of this particular 13-nt in length tRNA-fragment and showed that it is modulated by environmental stresses (nutritional and thermal stress signals) and is delivered through OMVs to human cells in culture promoting host cell gene expression and proliferation. The authors also started to explore the molecular mechanisms of these regulatory effects and suggested the competition with gene silencing properties of miRNA.

Overall, the paper is well written and the experiments are well performed. However, the work is rather descriptive and largely based on another submitted paper of the same team. Novelty of the present work as compared to previous studies implicating tRNA-fragments in MAPK-signalling pathways should be emphasized.

I have several concerns that should be addressed prior to publication.

Reviewer #2: Diallo and colleagues have investigated the role of a very small bacterial RNA (vsRNA) produced by the commensal bacterium E. coli in the context of bacteria- host interaction. This vsRNA named Ile-tRF-5X originates from the fragmentation of a tRNA, Ile-tRNA, and is transported by OMVs produced by E. coli MG1655. Specifically, they have shown that Ile-tRF-5X production can be modulated by bacterial growth conditions but not by inhibition of transcription and translation and that RNaseE is involved in its production. Ile-tRF-5X can be transferred via OMVs to HCT116 human cells where it remains relatively stable. The abundance of MAP3K4 mRNA that contain several Ile-tRF-5X binding sites is induced by Ile-tRF-5X, while OMV that contain Ile-tRF-5X also induce MAP3K4 at the protein level. The authors also evaluated the impact of the miRNA pathway on the Ile-tRF-5X-dependent modulation of MAP3K4 expression to decipher the underlying regulatory mechanism. Finally, the impact of OMV and Ile-tRF-5X on HCT116 cell proliferation and expression of various genes involved in inflammation and apoptosis have been investigated. Since the role of OMV and its RNA cargo in mediating host microbiota interactions is a relatively new area, this manuscript makes a meaningful contribution to the field that is likely to be of broad interest. Overall, the manuscript is well written, the experiments sound, and the conclusions reasonable. However, there are some areas where the manuscript should be improved before publication, including additional controls in experiments.

**Part II – Major Issues: Key Experiments Required for Acceptance**

Reviewer #1: 1. Previous results are cited several times in the manuscript as (Diallo et al submitted), however, the submitted paper is not available. It is not easy to follow the current manuscript without these data. For example, page 5 « are predicted to target several host mRNAs with diverse function (Diallo et al, submitted) », please provide details. Page 12, “several tRNAs might be targeted by tRF vsRNAs (Diallo et al, submitted)”.

I wondered if authors considered the BioRxiv submission of the cited paper ?

2. The generation of tRNA-derived fragments has been reported to impact cell proliferation through MAPK-signalling pathway modulation in previous work on OMV-associated short RNAs. The novelty of the present study should be emphasized as compared to previously reported data.

3. I wondered how the bacterial origin of the tRNA fragments could be proven? Are there similar fragments in human cells? Please consider this point.

4. The level of Ile-tRF-5X should be compared with the level of precursor/mature Ile-tRNA. The normalization strategy should be specified.

5. Page 9: Whether the effects of additional ribonucleases has been assessed to get insight into the generation of tRF?

6. Page 27: It will be interesting to explore the impact of endogenous tRF on MAPK expression and proliferation as compared to OMVs-derived tRF investigated in this study. The possible interference between endogenous tRF and OMVs-derived tRF should be considered.

Reviewer #2: Major comments:

- In many place in the MS, the authors refer to data obtained in a previous work and submitted recently to justify some experiments and/or to strengthen their results. For example:

P5: “The loading of E. coli vsRNAs, especially tRFs, in OMVs seemed to be selective. We also observed that tRFs are probably produced upon specific processing of tRNA species, form thermodynamically stable hairpin structures, and are predicted to target several host mRNAs with diverse functions (Diallo et al., manuscript submitted).”

P6: “In a rich and complete LB medium, E. coli Ile-tRF-5X level

seemed not to be affected by the growth phase, as suggested by our RNA-Seq data (see

Supplementary Figure S3). »

P8:” These RT-qPCR results confirm our RNA-Seq data suggesting that the level of bacterial Ile-tRF-5X is not affected by inhibition of transcription or translation (Supplementary Figure S3). »

P10:” We have previously shown that a group of thermodynamically stable bacterial vsRNAs, including Ile-tRF-5X (Figure 1), were selectively enriched and loaded into OMVs (Diallo et al., manuscript submitted; Supplementary Figure S4), suggesting a potential role in bacteria-host cell communications. “

It is difficult for the reviewer to deal with this information provided by the authors as a fact while it has not been still reviewed elsewhere. At times, the results also appear as validations of previous results to which we do not yet have access.

- The aim of the present study is to demonstrate that Ile-tRF-5X which is transported by OMV modulates host response. The authors have clearly demonstrated the modulatory role of Ile-tRF-5X in the host using a transfection approach. However, when OMVs are used, the vehicle carrying Ile-tRF-5X, the results are sometimes different. This is understandable since OMVs contain many other molecules than Ile-tRF-5X such as other RNAs and also proteins... Therefore, it is not clear whether Ile-tRF-5X when present in OMV are able to modulate host response, the aim of the study. To demonstrate the regulatory role of Ile-tRF-5X contained in OMVs it would be necessary to compare WT OMVs with OMVs lacking Ile-tRF-5X. For that an E. coli mutant lacking Ile-tRF-5X or ILE-tRNA should be used. Such a mutant strain is probably not possible to obtain as discussed by the authors. Nevertheless, one can suppose that OMVs produced by a RNaseE mutant, in which the level of ILE-tRNA is negatively affected could be used as OMVs with a decreased ILE-tRNA content. In addition, the authors could also use OMV-derived sRNA and OMVs treated with Ile-tRF-5X antisens, notably in experiments corresponding to figures 10/13/14 and S8.

- It is not obvious to understand what controls have been used in some experiments, and how data have been normalized (eg reference condition). Moreover the way of representing some results (relative data vs. raw data) is not always appropriate.(see minor comments for more details)

- The discussion section is very long and deals with elements that are not directly related to the current work (see minor comments for more details).

**Part III – Minor Issues: Editorial and Data Presentation Modifications**

Reviewer #1: - The total number of figures is too much (15 main figures). Some of them should be combined as panels of the same figure or moved to the supplementary material.

- Figures but not their titles and legends are included in the main text, please correct.

- Page 3 : « For instance » should be removed.

- Page 4. « Bacterial OMVs are produced and released in all domains of life » please check, seems contradictory « bacterial »/ “all domains of life “

- Page 4 Escherichia coli (E. coli) : « (E. coli) » to remove, please indicate the full name the first time and the « E. coli » from the second time it appears in the text.

- Page 4 : « five other bacterial strains » please specify which strains ?

- Page 4 « Gram-positive »

- Page 5 Not clear for the name of Ile-tRF-5X

- Page 6 I wondered whether 2-fold modulation observed is sufficient to have an impact on further regulatory processes?

- Page 8 “RT-qPCR (data not shown)”. It is important to show the data. Please specify “tRNA” in general or Ile-tRNA ?

- Page 9: On my opinion it is difficult to conclude that “Ile-tRF-5X is a product of a specific process” without providing additional mechanistic details

- Page 10: The choice of cell line/markers should be explained.

- Page 11: The differences in the delivery for OMVs-associated and synthetic Ile-tRF-5X could affect the observations, please emphasize this point.

- Figure 10B: Significant effect of synthetic Ile-tRF-5X in Dicer-deleted background?

- Page 19/25: “Dicer-derived miRNA” please specify which miRNA?

- Page 26: “contribute to stabilize” please check “contribute to stabilization”, however, the stability of MAP3K4 has not been assessed.

- Figure 15: This is a general figure more suitable for a review article. It should be removed or replaced by a schematic view of the model for Ile-tRF-5X effects.

- Page 34 “strategy)” bracket to remove

- Page 36/38 GEO accession number to specify

- Page 38 “Pseudomonas strains” Please check. It is E. coli that was used in this paper if I am right.

- Page 43 References: the species names should be in italic

- Figure S3. Only one biological replicate is presented, please check?

Reviewer #2: Minors comment

- Page 4: Only one example of trans-kingdom activity of regulatory sRNAs contained in OMVs is provided in the introduction section (Pseudomonas spp). Since the scientific literature is scarce in the field, the authors should provide more examples, notably introduce the recent work of Sahr and colleagues (DOI: 10.1038/s41467-022-28454-x) that studied the role of tRNA-Phe contained in OMVs.

- Figure2A. Please indicate whether the Ile-tRF-5X level is significantly or not different in M63 Expo vs Stat. In the discussion section (page 22), it is mentioned that the Ile-tRF-5X level remains unchanged during the Expo or Stat phase. Results presented in figure 2 seems to indicate otherwise.

- Evaluation of Ile-tRF-5X expression under different conditions (Figure2-3-4). Ile-tRF-5X expression was evaluated by LNA RT-qPCR. It is mentioned in the Figure 2 caption and M&M section that UniSp6 was used as spike-in control for cDNA synthesis and PCR amplification as well as reference gene for normalization. For gene expression normalization in RT-qPCR we used internal genes known to be (or supposed to be) constitutively expressed in the tested conditions. Spike-ins can be used to control RNA isolation and/or to monitor cDNA synthesis and/or amplification efficiency. I do not understand how this exogeneous spike-in “gene” can be used as an internal reference gene. Please clarify this crucial point.

- P9, analysis of Ile-tRF-5X expression according to RNaseE/P availability : The quantification of Ile-tRF-5X expression in the RNaseE mutant measured at 44°C is compared to the level of Ile-tRF-5X at 37°C (reference condition). In this context, the reference condition should be 44°C??? Please comment. What is known about the modulation of RNAseE activity by environmental conditions such as LB vs M63 and 37 vs 44°C? To make a link between the modulation of Ile-tRF-5x amount and RNaseE activity it would be interesting to determine Ile-tRF-5x expression in M63 medium in the WT and RNasE mutant strain.

- Bacterial OMV and Ile-tRF-5X. It is mentioned that MG1665-derived OMVs contain Ile-tRF-5x. However, this information is not shown. It is shown that Ile-tRF-5X level is modulated by environmental conditions encountered by the bacterium. It would also have been interesting to determine whether the level of OMV-derived Ile-tRF-5X is also modulated by environmental conditions.

- P12 “Mitogen-activated protein kinase 3, which carries no less than 10 potential binding sites”. Is the protein, the gene or the mRNA that contains the Ile-tRF-5X binding sites??? Please clarify the sentence.

- Figure 7. It is mentioned that the size of the seed sequence ranked as 1 on MAP3K3 RNA (ie the Ile-tRF-5X binding site) is a 13-mer. Why only a 12-mer sequence is reported on the corresponding RNAhybrid box?

P14 “hinted to the presence of synergistic small RNAs”. At this stage, this part is very speculative and close to discussion. One can also suppose that the level of transfected Ile-tRF-5X is lower compared to the Ile-tRF-5X amount associated to OMV. Please modify the sentence.

P14: I found no information about the mutated version of Ile-tRF-5X in the MS.

-Figure 8 and 9. A variability of the impact of Ile-tRF-5X on MAP3K4 expression can be observed between experiments: 6 fold induction in figure 9A vs 3-fold induction in figure 9C. It would have been interesting to compare the impact of OMV, OMVs-derived sRNAs and Ile-tRF-5X on MAP3K4 expression from the same experiments.

- P15 and figure 10: it is postulated that MAP3K4 expression is downregulated by miRNAs and that Ile-tRF-5X competes with miRNAs to upregulate MAP3K4. To demonstrate this hypothesis, the effect of Ile-tRF-5X on MAP3K4 expression was evaluated in a cell line (Dicer -/- ) in which miRNAs are mostly depleted and compared to the wild type cell line. MAP3K4 expression was evaluated both in WT and Dicer strains but from figure 10B we can not directly compared the result of this comparative analysis since data were normalized to mock. Indeed, if the hypothesis is correct the expression of MAP3K4 should be higher in the Dicer strain even in absence of Ile-tRF-5X. In addition, the authors showed that the positive impact of OMV and Ile-tRF-5X on MAP3K4 is reduced in the Dicer strain. This result does not support the initial hypothesis on a competition between miRNA and Ile-tRF-5X… Moreover, this result is contradictory with that presented in Figure 12C, which shows a loss of the inhibition of the compared Rluc reporter expression in the Dicer strain (100 % RLuc) when compared to the WT strain (40 % RLuc). Please clarify these apparent inconsistencies. Finally I am wondering why the authors used OMV instead OMV-derived sRNAs in this experiment as OMVs and not OMV-derived sRNAs as a positive effect on the MAP3K4 protein level?

-Figure 7 and Figure 11. Why are none of the Ile-tRF-5X binding sites shown in Figure 11 present in Figure 7?

- Figure8, 9, and 10. Please indicate in figure captions the quantity of OMV and Ile-tRF-5X used in the different experiments (as reported for figure12 and 13).

-Figure 12AB and page 18: The author mentioned that the Rluc activity is less reduced when Ile-tRF-5X binding sites are mutated (35% of reduction) when compared to WT Ile-tRF-5X binding sites (60%). Are these differences really significant? If no, it seems difficult to suggest that mutations in Ile-tRF-5X binding sites affect the inhibitory elements of the 3’UTR as indicated by the authors. Page19 it is mentioned that “ Ile-tRF-5X was still able to restore Rluc activity, but at a reduced intensity…”. The Rluc activity in presence of Ile-tRF-5X (50 ng) appears similar with the WT (90%, Fig12A) and the mutated Ile-tRF-5X binding sites (90%, Fig12B) and not reduced when Ile-tRF-5X binding sites are mutated. Please correct. Moreover, how can we explain that Ile-tRF-5X (50 ng) still continues to activate the Rluc activity when Ile-tRF-5X binding sites are mutated (Fig12B)?

- Figure 13. Why the cell proliferation level is 6-fold higher in absence of OMV (panel A, point 0) than in absence of Ile-tRF-5X (panel B, point 0)? This is the same condition? It is mentioned in the figure caption that data were normalized with its corresponding control. The controls are similar or different between the two panels? (which are the control conditions?). Moreover, why two different statistical tests were applied between panel A (two way ANOVA) and B (one way ANOVA) with different post-hoc test, while their corresponding experimental design looks similar?

I am not sure to really understand the experiment and the format used to present the data obtained is confusing. The XTT is used to quantify viable cells. Therefore, for example, 3.E3 cells are found viable after 48 h of incubation with the control, while 10.E3 cells are viable after 48 h of incubation in presence of 10 ng of OMV (Figure13 panelA). These raw data can be compared if the initial number of cells is strictly the same (this is rarely the case, since there is some variability from one well to another within a plate in cellular culture). To overcome this, the data are often expressed in a relative way by dividing the number of cells obtained by the number of initial cells and expressed as a percentage, eg 100% viable cells after 48h in the control conditions, vs 200% after 48h in presence of OMV.

Since it has been shown that OMV and sRNA OMV can have a different impact on the MAPK pathways (Figure 8 vs Figure 9), an antisense Ile-tRF-5X has no impact on OMV-induced cell proliferation (Fig S7), and OMV can induce cell mortality (but not Ile-tRF-5X), It would be interesting and necessary to see and compare the effect of OMV-derived sRNA on cell proliferation. Finally, why the authors decided to measure cell proliferation after 48h of interaction (incubation). This time is surprising as they showed a drastic drop (74%) of Ile-tRF-5X level in HCT116 cells at 48 h when compared to its level at 24h (figure 6)?

- Figure 14. The ability to induce the expression of various HCT116 genes differs between OMVs and Ile-tRF-5X. To claim that “bacterial OMVs and their Ile-tRF-5X content are

involved in the controlled regulation of human cell proliferation and death”, It should also have been determined the effect of OMV-derived sRNA OMVs on gene expression, notably Casp-1, Bax and Bcl-2. In addition, Ile-tRF-5X induces MAP3K4 expression without inducing the upregulation of MAP3K4 at the protein level. In this context, how to explain the impact of Ile-tRF-5X on P38a expression?

- Page 22: “However, under normal growth conditions, either during…”. Can we consider that a growth in LB medium corresponds to a normal growth condition for an intestinal bacterium? I think the most appropriate term is “normal laboratory conditions”

-Page 24: “The selective enrichment of vsRNAs in OMVs and their effective transfer to human host cells were previously evidenced by other teams (34,40,81,82).” In the present MS, the authors never show an enrichment of RNA into OMV. This part is not appropriate. In addition, the comparison of the effect of OMV in conventionalized vs germ-free mice is out of the scope of the MS.

-Page 25: I am not sure that the potential adjuvant effect of OMV can be an argument to explain why OMV can induce MAP3K4 at the protein level protein and not it sRNA content…

- Page 27: “which is required for cell proliferation and survival…”. It means that Ile-tRF-5X displays proto-oncogene activity. I think this part of the discussion (including the end) should be weighted. Indeed, the data generated were obtained from a colon cancer cell line.

PLOS authors have the option to publish the peer review history of their article (what does this mean?). If published, this will include your full peer review and any attached files.

Reviewer #1: No

Reviewer #2: **Yes: **Eric Guédon
---

## [Decision Letter · Decision Letter 1]

22 Jul 2022

Dear Dr. Provost,

Thank you very much for submitting your manuscript "A tRNA-derived fragment present in E. coli OMVs regulates host cell gene expression and proliferation" for consideration at PLOS Pathogens. As with all papers reviewed by the journal, your manuscript was reviewed by members of the editorial board and by several independent reviewers. The reviewers appreciated the attention to an important topic. Based on the reviews, we are likely to accept this manuscript for publication, providing that you modify the manuscript according to the review recommendations.

Please carefully read over the manuscript and modify the text as requested by the referees. Then, prepare and submit your revised manuscript within 30 days. If you anticipate any delay, please let us know the expected resubmission date by replying to this email.

Sincerely,

Mathieu Coureuil

Guest Editor

PLOS Pathogens

Xavier Nassif

Section Editor

PLOS Pathogens

Kasturi Haldar

Editor-in-Chief

PLOS Pathogens

orcid.org/0000-0001-5065-158X

Michael Malim

Editor-in-Chief

PLOS Pathogens

orcid.org/0000-0002-7699-2064

Reviewer Comments (if any, and for reference):

Reviewer's Responses to Questions

**Part I - Summary**

Reviewer #1: The authors responded to the major comments and modified the manuscript accordingly.

Reviewer #2: The authors have responded point by point to all my previous remarks and modified the text accordingly. I strongly recommend the publication of the present MS in the PLOS pathogens journal.

**Part II – Major Issues: Key Experiments Required for Acceptance**

Reviewer #1: The major concern was the availability of the previous data cited in the manuscript. The paper has been now published in Frontiers in Mol Biosciences and the authors included this citation throughout the manuscript.

Reviewer #2: No new experiments are required

**Part III – Minor Issues: Editorial and Data Presentation Modifications**

Reviewer #1: The minor concerns have been mainly addressed. Few points like the names of the bacterial species in italic in the text need to be checked in final version. Some remaining typos should be corrected like Fig 4 legend "Bacteria RNase E" to be changed to "Bacterial RNase E" etc.

Reviewer #2: Page 2: abstract section : Check the bracket in italic after E. coli : (E. coli)

Page 4: Please check “Bacterial Extracellular vesicles”

Page 4: “L. pneumophila”. For the first time, please mention the species with its full name and in italic.

Page 5: “Pseudomonas… HG001”, please check the typo.

Page5: “These vsRNAs were highly abundant in bacteria and their derived OMVs”. It seems to me that this has only been demonstrated for E. coli derived OMV… Please modify the sentence.

Figure 1 caption: Please modify “isoleucin” by isoleucine and indicate genes in italic and without upper case for the first letter.

Page 7 (lane2): E. coli in italic.

Page 13: Please change Figure S54 by Figure S5. In addition, in Figure S5 caption two panels are mentioned (A for 2h and B for 18h). However, only one panel is shown in Fig. S5. Please modify.

Figure 6 legend: check “withE. coli”. Please also indicate the quantity of OMV and synthetic Ile-trf-5X used like in Figure 7 and 8

Figure 7 legend: change “10ng” by “10 ng”

Page 17: I didn’t find in the MS any information about the mutated Ile-tRF-5X (sequence, ..)

Figure 9 legend: please modify “10ng” and “100nM”

Page 25: Check the double bracket after reference 60

Page 25: please modify “may seems counterintuitive”

PLOS authors have the option to publish the peer review history of their article (what does this mean?). If published, this will include your full peer review and any attached files.

Reviewer #1: No

Reviewer #2: **Yes: **Éric GUÉDON

Figure Files:

Data Requirements:

Reproducibility:

References:

---

## [Editor Report · Decision Letter 2]

22 Aug 2022

Dear Dr. Provost,

We are pleased to inform you that your manuscript 'A tRNA-derived fragment present in E. coli OMVs regulates host cell gene expression and proliferation' has been provisionally accepted for publication in PLOS Pathogens.

Best regards,

Mathieu Coureuil

Guest Editor

PLOS Pathogens

Xavier Nassif

Section Editor

PLOS Pathogens

Kasturi Haldar

Editor-in-Chief

PLOS Pathogens

orcid.org/0000-0001-5065-158X

Michael Malim

Editor-in-Chief

PLOS Pathogens

orcid.org/0000-0002-7699-2064
---

## [Editor Report · Acceptance letter]

2 Sep 2022

Dear Dr. Provost,

We are delighted to inform you that your manuscript, "A tRNA-derived fragment present in *E. coli* OMVs regulates host cell gene expression and proliferation," has been formally accepted for publication in PLOS Pathogens.

Best regards,

Kasturi Haldar

Editor-in-Chief

PLOS Pathogens

orcid.org/0000-0001-5065-158X

Michael Malim

Editor-in-Chief

PLOS Pathogens

orcid.org/0000-0002-7699-2064